# Resolution of Simpson's paradox via the common cause principle

**Arshak Hovhannisyan** [*]
Alikhanyan National Laboratory, Yerevan, Armenia

**Armen Allahverdyan** [†]
Alikhanyan National Laboratory, Yerevan, Armenia

## Abstract

Simpson's paradox poses a challenge in probabilistic inference and decision-making. Our study revisits the paradox by re-estimating its frequency with an unbiased data generation process and reaffirms that it is not an artifact of deficient data collection. Thus, it can lead to incorrect recommendations in fields as diverse as statistics, psychology, and artificial intelligence. We show that the paradox can be resolved by assuming a minimal — though not necessarily observed — common cause (or screening) variable for the involved random variables. In our approach, conditioning on this minimal common cause establishes the correct association between events, which coincides with the conditioning (i.e., fine-grained) option of the original Simpson paradox. This resolution applies to both discrete cases of binary variables and continuous settings modeled by Gaussian variables. For a non-minimal common cause, the resolution of the paradox is possible, but detailed knowledge of the common cause is required. Our findings extend traditional understandings of the paradox and offer practical guidance for resolving apparent contradictions in probabilistic inference, ultimately enhancing decision-making processes. This point is illustrated by several examples.

## 1 Introduction

Simpson's paradox was discovered more than a century ago [1, 2], generated a vast literature, and is well-recognized in several fields including, statistics, epidemiology, psychology, social science, *etc.* [3–24]. This counter-intuitive effect limits the ability to draw conclusions from probabilistic data. The effect is important because it demands more than simply extracting relative frequencies from data; e.g. it necessitates looking at exchangeability [9] or causality [7–9, 13, 14].

The paradox starts with two random variables $A$ and $B$. Now $A = (A_1, A_2)$ contains control variable $A_2$ and the target variable $A_1$, while $B$ is a side random variable that depends on both $A_1$ and $A_2$. The meaning of $A$ and $B$ is clarified via examples presented below. If there is no information on the outcome of $B$, the behavior of $A$ can be studied on two levels. The first (aggregated) level is that of marginal probabilities $p(A = a)$. The second level is finer-grained and is represented by conditional probabilities $p(A = a | B = b)$ for all possible values of $B$. Simpson's paradox amounts to certain relations between those probabilities; see section 2 for details. It states that no decision-making is possible, because conclusions drawn from probabilities on different levels contradict each other. Without Simpson's paradox, decision-making can proceed at the aggregate level, because looking at the fine-grained level is either redundant or inconclusive. Thus, Simpson's paradox first and foremost

---

[*]arshak.hovhannisyan@aanl.am

[†]a.allahverdyan@aanl.am

39th Conference on Neural Information Processing Systems (NeurIPS 2025).

involves decision-making. Moreover, it demonstrates limitations of the sure-thing principle [5], a pillar of traditional decision making [25–27]. A recent review of the sure-thing principle (and its limitations other than Simpson's paradox) can be found in Ref. [28]. Limitations of probabilistic decision-making are important for the modern artificial intelligence (probability models, uncertainty estimation, *etc*).

In section 2, Simpson's paradox is defined in detail, and previous efforts to resolve it in several specific situations are reviewed and criticized. In particular, we show that while certain previous solutions of the paradox assumed the existence of (causally-sufficient) time-ordered directed acyclic graphs (TODAGs) that describe the 3 variables involved in the paradox, several important examples of the paradox need not support this assumption; see sections 2.2.2, 4 and 5. Based on the previous literature, we argue that Simpson's paradox is sufficiently frequent when the probabilities of the involved variables are generated from the unbiased (non-informative) distribution, modeled via Dirichlet density; see Appendix A. Hence this is a genuine decision-making paradox and not an artifact due to inappropriate data gathering.

Our proposal here is to search for the resolution of the paradox by assuming - given two dependent random variables $A$ and $B$ - there is a random variable $C$ that makes $A$ and $B$ conditionally independent; i.e., screens out $A$ from $B$. Examples of Simpson's paradox show that such a $C$ is frequently plausible, though it is normally not observed directly. In particular, $C$ is conceivable if the dependence between $A$ and $B$ are not caused by a direct causal influence of $B$ on $A$. Then the existence of $C$ is postulated by the common cause principle. (If the dependence is due to a causal influence of $A$ on $B$, Simpson's paradox can formally exist, but factually it is absent because the decision is obviously to be taken according to the aggregated level.)

Introducing the screening variable $C$ allows us to reformulate and extend Simpson's paradox: its two options - along with many other options - refer to particular choices of $C$; see section 3. Now, the paradox seems to be further from being resolved than before. However, we show that when the variables $A_1$, $A_2$, $B$, and $C$ holding the paradox are binary (the minimal set-up of the paradox), the decision-making is to be made according to the fine-grained probabilities, i.e., the paradox is resolved. Such a definite relation is impossible for a tertiary (or larger) $C$: now depending on $C$ all options of Simpson's paradox are possible, e.g. the precise control of $C$ can be necessary for decision-making.

Next, we turn to Simpson's paradox for continuous variables, which was discussed earlier than the discrete formulation [1]. It holds the main message of the discrete formulation. In addition, it includes the concept of the conditional correlation coefficient (only for Gaussian variables is the random-variable dependence fully explained by the correlation coefficient). The continuous formulation is important because it applies to big data [23, 24, 29], and because (statistically) it is more frequent than the discrete version [30]. The advantage of continuous Gaussian formulation is that the general description of the paradox under the common cause is feasible; see section 6. For this situation, we show conceptually the same result as for the discrete version: in the minimal (and most widespread) version of the paradox, the very existence of an (unobservable) common cause leads to preferring the fine-grained option of the paradox.

The rest of this paper is organized as follows. Section 2 is a short but sufficiently inclusive review of Simpson's paradox and its resolutions proposed in the literature [3]. It also discusses two basic examples for illustrating different aspects of the paradox; see section 2.2.2. In section 3 we reformulate Simpson's paradox by assuming that there is a common cause (or screening variable) $C$ behind the three variables. Now $C$ need not be observable, since we show that it will be sufficient to assume that it exists and (provided that all variables are binary) Simpson's paradox is resolved by choosing its fine-grained option. A similar conclusion is reached for Gaussian variables; see section 6. Section 4 considers published data from Ref. [16] on a case of smoking and surviving. This example is not easily treated via the existing methods. Still, we show that the existence of a common cause for this situation is plausible and that Simpson's paradox can be studied via our method and leads to a reasonable result. Section 5 treats data on COVID-19, which was suggested in Ref. [31]. We demonstrate that an assumption of a plausible common cause points to different conclusions than in Ref. [31]. The last section summarizes our results and their limitations. It also outlines future research directions.

---

[3]Among the issues not addressed in this paper is the explanation of Simpson's paradox using counterfactual random variables. This subject is reviewed in [6].

## 2 Formulation of Simpson's paradox and previous works

### 2.1 Formulation of the paradox for binary variables and its necessary conditions

To formulate the paradox in its simplest form, assume three binary random variables $A_1 = \{a_1, \bar{a}_1\}$, $A_2 = \{a_2, \bar{a}_2\}$, $B = \{b, \bar{b}\}$. The target event is $a_1$, and we would like to know how it is influenced by $A_2$ which occurs at an earlier time than the time of $A_1$: $t_{A_2} \leq t_{A_1}$. This can be done by looking at conditional probability. For

$$p(a_1|a_2) < p(a_1|\bar{a}_2), \tag{1}$$

which is equivalent to $p(a_1) < p(a_1|\bar{a}_2)$, we would conclude that $\bar{a}_2$ enables $a_1$. However, (1) is compatible with

$$p(a_1|a_2, b) \quad > p(a_1|\bar{a}_2, b), \tag{2}$$
$$p(a_1|a_2, \bar{b}) \quad > p(a_1|\bar{a}_2, \bar{b}), \tag{3}$$

where $B$ also occured in an earlier time: $t_B \leq t_{A_1}$. Examples supporting (1–3) are studied below (sections 2.2.2, 4 and 5) and also Appendix C. Since (2, 3) hold for each value of $B$ we should perhaps conclude that $a_2$ enables $a_1$ in contrast to (1). Decision-makers would not know whether to apply (1) or (2, 3). This is Simpson's paradox. Its equivalent formulation is when all inequalities in (1–3) are inverted [4].

For Simpson's paradox (1–3) to hold, it is necessary to have one of the following two conditions:

$$p(a_1|\bar{a}_2, b) < p(a_1|a_2, b) < p(a_1|\bar{a}_2, \bar{b}) < p(a_1|a_2, \bar{b}), \tag{4}$$
$$p(a_1|\bar{a}_2, \bar{b}) < p(a_1|a_2, \bar{b}) < p(a_1|\bar{a}_2, b) < p(a_1|a_2, b). \tag{5}$$

To find these relations, expand $p(a_1|a_2)$ and $p(a_1|\bar{a}_2)$ over the probabilities in (4, 5) [cf. (56, 57)], and note that e.g. $p(a_1|a_2)$ is a weighted mean of $p(a_1|a_2, b)$ and $p(a_1|a_2, \bar{b})$. Given (4) or (5), Simpson's paradox can be generated via suitable choices of $p(b|a_2)$ and $p(b|\bar{a}_2)$; see Appendix A.

### 2.2 Attempts to resolve the paradox

#### 2.2.1 Replacing prediction with retrodiction

Over time, several resolutions to the paradox have been proposed. Barigelli and Scozzafava [10, 11] proposed to replace (1) by

$$p(a_2)p(a_1|a_2) < p(\bar{a}_2)p(a_1|\bar{a}_2), \tag{6}$$

i.e. to interchange $A_1$ and $A_2$ in (1). Then it is easy to see that its inversion under additional conditioning over $B$ is impossible. While (1) stands for prediction – i.e. aiming at $a_2$ (and not at $\bar{a}_2$) will more likely produce $\bar{a}_1$ (than $a_1$) – the proposal by Ref. [10, 11] looks for retrodiction. Though retrodicting (in contrast to predicting) does not suffer from Simpson's paradox, retrodicting and predicting are different things, and cannot generally be substituted for each other.

Rudas also sought to change the criterion (1) so that it does not allow inversion after additional conditioning over $B$, but still has several reasonable features [32]. The proposal is to employ $p(a_2)[p(a_1|a_2) - p(\bar{a}_1|a_2)] < p(\bar{a}_2)[p(a_1|\bar{a}_2) - p(\bar{a}_1|\bar{a}_2)]$ instead of (1) [32]. Notice the conceptual relation of this with the previous proposal (6).

An unnatural point of both these proposals is that they depend on the ratio $p(a_2)/p(\bar{a}_2)$; e.g. for the **Example 1** mentioned below this means that if the treatment was applied more, it has better chances to be accepted. This drawback is acknowledged in [32].

#### 2.2.2 Exchangeability and causality

According to Lindley and Novick, the paradox may be resolved by going beyond probabilistic considerations (as we do below as well) and by employing the notion of exchangeability or causality [9]; see

---

[4]We leave aside the following pertinent problem; see [19] for details. If probabilities are extracted from finite populations, the more conditioned version (2, 3) is less reliable, because it is extracted from a smaller population. For us all probability-providing populations will be sufficiently large.

[33] for a recent discussion on causality and exchangeability. Within that proposal, the data generally provides only propensities, and one needs additional assumptions of sample homogeneity (exchangeability) for equating propensities with probabilities *even* for a large sample size. Exchangeability and the closely related notion of ergodicity remain influential in the current analysis of statistical problems exemplified by Simpson's paradox [34]. Lindley and Novick studied the following two examples that support Simpson's paradox (more examples are discussed in sections 4, 5, and Appendix C).

**Example 1.** Medical treatment [9]. $A_1 = \{a_1, \bar{a}_1\}$ (the target variable) is the recovery rate of medical patients: $a_1 = $ recovery, $\bar{a}_1 = $ no recovery. $A_2 = \{a_2, \bar{a}_2\}$ refers to a specific medical treatment: $a_2 = $ treatment, $\bar{a}_2 = $ no treatment. $B = \{b, \bar{b}\}$ is the sex of patients: $b = $ male, $\bar{b} = $ female. The times to which the random variables $A_1$, $A_2$ and $B$ refer clearly hold $t_B < t_{A_2} < t_{A_1}$.

**Example 2.** Plant yield [9]. $A_1 = \{a_1, \bar{a}_1\}$ (the target variable) is the yield of a single plant: $a_1 = $ high, $\bar{a}_1 = $ low. $A_2 = \{a_2, \bar{a}_2\}$ refers to the variety (color) of the plant: $a_2 = $ dark, $\bar{a}_2 = $ light. $B = \{b, \bar{b}\}$ refers to the height of the plant: $b_1 = $ tall, $\bar{b} = $ low. The times hold $t_{A_2} < t_B < t_{A_1}$.

Lindley and Novick proposed that assumptions on exchangeability lead to preferring (1) for **Example 2** and (2, 3) for **Example 1** [9]. They also proposed that the same results can be found by using causality instead of exchangeability [9]. The same proposal was made earlier by Cartwright in the context of abstract causality [7, 8]. Pearl elaborated this proposal assuming that the above examples can be represented via time-ordered direct acyclic graphs (TODAG) [13, 14], where an arrow $\rightarrow$ represents the influence of an earlier variable to the later one; see Fig. 1 for details. If we follow this assumption, then - given the time constraints for the examples - each of them can be related to a unique TODAG:

$$\text{Example 1}: \quad B \rightarrow A_2 \rightarrow A_1 \leftarrow B, \tag{7}$$
$$\text{Example 2}: \quad A_2 \rightarrow B \rightarrow A_1 \leftarrow A_2. \tag{8}$$

In (7) the suggestion is to condition over $B$ [hence using (2, 3)] if $B$ influences both $A_1$ and $A_2$ [9, 13, 14]. This is because conditioning over the cause reduces spurious dependencies. This reasoning was generalized as the back-door criterion [13]. In contrast, it is advised to use (1) in (8) since $B$ is an effect of $A_2$, but still a cause of $A_1$ [9, 13, 14]. The intuition of this suggestion is seen in the extreme case when $B$ screens $A_1$ and $A_2$ from each other, i.e. $A_1$, $B$ and $A_2$ form a Markov chain. Then the conditional probability $p(A_1|A_2, B) = p(A_1|B)$ will not depend on $A_2$ begging the original question in (1). Thus, for the two examples considered in [9], Refs. [13, 14] make similar recommendations. The basis of these recommendations was criticized in [17].

Refs. [13, 14] imply that Simpson's paradox for (7, 8) can be solved via do-calculus. This is only partially correct: only (7) is solved with do-calculus. Indeed, the do-calculus in (7) defines $p(a_1|\text{do}(A_2)) = \sum_b p(b)p(a_1|A_2, b)$. Now $p(a_1|\text{do}(A_2 = a_2)) > p(a_1|\text{do}(A_2 = \bar{a}_2))$, amounts to the fine-grained version (2, 3) of the paradox, which agrees with the conclusion of [9]. However, for (8) we have $p(a_1|\text{do}(A_2)) = p(a_1|A_2)$, and it is not clear what prevents us from going back to the original formulation of Simpson's paradox, i.e., comparing $p(a_1|\text{do}(A_2 = a_2)) < p(a_1|\text{do}(A_2 = \bar{a}_2))$, with $p(a_1|\text{do}(A_2 = a_2), b) > p(a_1|\text{do}(A_2 = \bar{a}_2), b)$ and $p(a_1|\text{do}(A_2 = a_2), \bar{b}) > p(a_1|\text{do}(A_2 = \bar{a}_2), \bar{b})$.

Let us now argue that realistically, **Example 1** and **Example 2** need not to support TODAGs (7, 8), respectively. In fact, both arrows $B \rightarrow A_2$ and $B \rightarrow A_1$ in **Example 1** are generally questionable: sex need not influence the selection of the treatment, $B \not\rightarrow A_2$ (unless the data was collected in that specific way), and many treatments are sex-indifferent, i.e. $B \not\rightarrow A_1$. For **Example 1** it is more natural to assume that $B$ does not causally influence $A$. In such a situation, the common cause principle proposes that there is an unobserved random variable $C$, which is a common cause for $A$ and $B$ [35, 36]; see section 3. Similar reservations apply to **Example 2**: now $A_2 \rightarrow B$ is perhaps argued on the basis of color ($A_2$) being more directly related to the genotype of the plant, while the height ($B$) is a phenotypical feature. First, color-genotype and height-phenotype relations need not hold for all plants. Second (and more importantly), it is more natural to assume that the plant genotype influences both its color and height than that the color influences height. Hence the genotype can be a common cause for $A$ and $B$. Implications of such common cause scenarios are studied below.

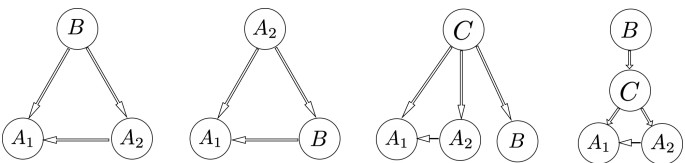

Figure 1: Directed acyclic graphs between random variables $A = (A_1, A_2)$, $B$ and $C$ involved in discussing Simpson's paradox. The first and second graphs were studied in Refs. [13, 14]; see (7, 8). The third or fourth graphs are basic assumptions of this work; see (9). In the first graph, $B$ influences $A_1$ and $A_2$, but $B$ is not the common cause in the strict sense, because there is an influence from $A_2$ to $A_1$. A similar interpretation applies to the second graph. We emphasize that the joint probability $p(A_1, A_2, B)$ for the first and second graphs has the same form, i.e. such graphs are extra constructions employed for interpretation of data. In contrast, the third and fourth graph imply a definite (but the same for both graphs) limitation on the joint probability $p(A_1, A_2, B, C)$, which is expressed by (9).

## 3 Common cause principle and reformulation of Simpson's paradox

### 3.1 Common cause and screening

The common cause for $A = (A_1, A_2)$ and $B$ means that there exists a random variable $C = \{c\}$ [35, 36]

$$p(A_1, A_2, B|C) = p(A_1, A_2|C)p(B|C), \tag{9}$$

$$p(A_1, A_2, B) = \sum_{c \in C} p(c)p(A_1, A_2|c)p(B|c), \quad p(c) > 0 \tag{10}$$

where (9) holds for all values assumed by $A_1$, $A_2$, $B$ and $C$, and where (10) follows from (9) [5]. The same (9) applies if $C$ causes $A$ and screens $A$ from $B$. These two scenarios are shown in Fig. 1 as (resp.) the third and fourth graphs. Sections 4, 5, and Appendix C provide several examples of a causing (or screening) variable $C$ in the context of Simpson's paradox. $p(A_1, A_2, B)$ in (10) can be considered as a matrix, where $(A_1, A_2)$ $[B]$ enumerates rows [columns]. Now there is a minimal value of $|C|$ (the number of realizations of $C$) such that (10) holds [37]. This minimal number is called the positive rank $(\mathrm{rank}_+[p])$ of the matrix $p(A_1, A_2, B)$ [3]. There are methods for its estimation [38]; e.g., it holds $\mathrm{rank}_+[p] \leq \min\left[|A_1| \cdot |A_2|, |B|\right]$, where $|B|$ is the number of realization of $B$. Representations (10) are not unique, even for a fixed $|C|$ [38].

The common cause principle was proposed to explain probabilistic dependencies [35, 36]. It later found important applications in data science, where approximate relations similar to (9) are applied to effective data compression (Non-negative matrix factorization, Probabilistic Latent Dirichlet indexing, *etc*) [39, 40]. Pearson and Yule expressed early ideas about the common causes that explain certain dependencies; see [41] for a historical review. Involving a common cause in Simpson's paradox means that we do not consider this paradox as referring to a causally sufficient situation; for recent discussions on causal (in)sufficiency see [42, 43].

Note from (9) that $C$ gets rid of the conditional dependence on $B$ in $p(A_1, A_2|B, C)$. Thus, a sensible way of looking at the association between $a_1$ and $a_2$ is to check the sign of

$$p(a_1|a_2, C) - p(a_1|\bar{a}_2, C) \quad \text{for each value of } C. \tag{11}$$

To support the usage of the common cause C for decision-making, we note that (9) has an important implication in the context of (1). (This implication generalizes the argument given in [36].) Assume that $p(a_2, b, c) > 0$ for all values $c$ of $C$. Note from (9) that there exists an event $c$ such that $p(a_1|a_2, b) \leq p(a_1|a_2, b, c) = p(a_1|a_2, c)$, and an event $c'$ such that $p(a_1|a_2, b) \geq p(a_1|a_2, b, c') = p(a_1|a_2, c')$. Hence, if conditioning over $b$ facilitates (hinders) the association between $a_1$ and $a_2$, then conditioning over $c$ $(c')$ is not worse in this facilitation (hindering) [6].

---

[5]There are formulations of the common cause principle that look for (9) holding for certain events only and not for random variables [35, 36]. We do not focus on them.

[6]To deduce the first relation assume that $p(a_1|a_2, b) < p(a_1|a_2, b, c) = p(a_1|a_2, c)$ for all $c$, multiply both parts by $p(a_2, b, c) > 0$, sum over $c$ and get contradiction $p(a_1, a_2, b) < p(a_1, a_2, b)$. Likewise for the second relation.

After the above reformulation, Simpson's paradox seems even less resolvable since $C$ is not observed. Indeed, there are common causes that reproduce (1), those that reproduce (2, 3), but there are many other possibilities. Common causes that are close to $B$ ($C \approx B$) imply option (2, 3) of the paradox, while $C \approx A$ leads to (1). These conclusions are based on the fact that (9) holds exactly for $C = B$ and $C = A$. Thus, Simpson's paradox is not a choice between two options (2, 3) and (1), it is a choice between many options given by different common causes $C$.

Finally, two remarks about the applicability of (9–11). First, if $C$ is a common cause for both $A = (A_1, A_2)$ and $B$, the times of these variables naturally hold $t_C < \min[t_{A_1}, t_{A_2}, t_B]$. When $C$ screens $A$ from $B$, it holds $t_B < t_C < \min[t_{A_1}, t_{A_2}]$. In certain applications of (11), it will suffice to have even a weaker condition $t_C < t_{A_1}$.

Second, we note that for applying (1, 2, 3) we do not need $p(A_2)$, i.e. only $p(B|A_2)$ is needed for connecting (1) with (2, 3). Indeed, $A_2$ does not necessarily need to be a random variable, but can simply be a label describing the situation. Now the same holds for (11): once (9) is written as

$$p(A_1, B|A_2, C) = p(A_1|A_2, C)p(B|C), \tag{12}$$

we need only $p(C|A_2)$ to pass from (12) to quantities involved in (1, 2, 3); i.e., $p(A_2)$ is not needed.

## 3.2   A common cause (or screening variable) resolves Simpson's paradox for binary variables

The following theorem shows a definite statement for all binary causes. The message of the theorem is that once we know that $C$ is binary, then the correct decision is (2, 3).

**Theorem 1:** If $A_1$, $A_2$, $B$ and $C = \{c, \bar{c}\}$ are binary, and provided that (1) and (2, 3) are valid, all causes $C$ hold

$$p(a_1|a_2, c) > p(a_1|\bar{a}_2, c), \qquad p(a_1|a_2, \bar{c}) > p(a_1|\bar{a}_2, \bar{c}), \tag{13}$$

i.e. all $C$ holding (9) predict the same sign of association between $a_1$ and $a_2$ as (2, 3). This theorem is proved in Appendix B. The main idea is that to prove (13), we need to invert (9).

The resolution of Simpson's paradox offered by Theorem 1 is consistent with the do-calculus; see [13] for a review. To show this, consider from Fig. 1 two TODAGs that support the causal structure of Theorem 1:

$$B \leftarrow C \rightarrow A_2 \rightarrow A_1, \qquad C \rightarrow A_1, \tag{14}$$
$$B \rightarrow C \rightarrow A_2 \rightarrow A_1, \qquad C \rightarrow A_1. \tag{15}$$

For both these TODAGs we have

$$p(a_1|\text{do}(A_2 = a_2)) = \sum\nolimits_{C=c,\bar{c}} p(a_1|a_2, C)p(C). \tag{16}$$

If (as stated in our theorem 1) $p(a_1|a_2, c) > p(a_1|\bar{a}_2, c)$ and $p(a_1|a_2, \bar{c}) > p(a_1|\bar{a}_2, \bar{c})$ then we get $p(a_1|\text{do}(A_2 = a_2)) > p(a_1|\text{do}(A_2 = \bar{a}_2))$. Hence, if the influence of $A_2$ on $a_1$ is decided via do-conditioning over $A_2$, then the conclusion agrees with the option (2, 3) of Simpson's paradox.

An important aspect of theorem 1, is that once we can motivate one of the above TODAGs (14, 15) with binary $C$, then no additional data-gathering is necessary for resolving the paradox, i.e., we do not need to know $P(A_1, A_2|C)$, *etc.*

Note the difference between TODAGs (7, 8) and (14, 15): (7, 8) are consistent with any joint probability $p(A_1, A_2, B)$. In contrast, (14, 15) require a specific probabilistic feature (9).

## 3.3   Non-binary causes

Let us assume that we have Simpson's paradox (1, 2, 3) and also the common cause condition (9). However, $C = \{c_1, c_2, c_3\}$ is now a tertiary random variable. It turns out that now all three options of Simpson's paradox become possible: there are common causes $C$ that support (1):

$$p(a_1|a_2, C) < p(a_1|\bar{a}_2, C), \text{ for } C = \{c_1, c_2, c_3\}. \tag{17}$$

There are also common causes $C$ which support (2, 3). Eventually, there are tertiary common causes $C = \{c_1, c_2, c_3\}$ for which $p(a_1|a_2, c_i) - p(a_1|\bar{a}_2, c_i)$ has different signs for different values of $i = 1, 2, 3$; i.e., neither (1), nor (2, 3) are supported. Hence, already for the tertiary cause, one needs

prior information on the common cause to decide on the solution of Simpson's paradox. Alternatively, we can infer this unknown cause via one of the methods proposed recently for obtaining the most plausible common cause [44, 45]. It is not excluded that such inference methods will provide further information on the solution of Simpson's paradox.

Note that (17) is a counter-example to an opinion that the structure of the TODAG as such can determine which option – (1) or (2, 3) – of Simpson's paradox applies. Indeed, (17) and **Theorem 1** support the same TODAGs (14, 15), but they lead to different options of the paradox.

## 4  Example: smoking and surviving

In section 2.2.2 we discussed two examples studied in the literature and argued that they can be also interpreted via the common cause principle. In the present case, the standard approaches do not seem to apply, but the common cause can still be motivated. This example on survival of smokers *versus* nonsmokers is taken from Ref. [16]. Its technical details are discussed in Appendix D. Binary $A_1$ represents the survival in a group of women as determined by two surveys taken 20 years apart:

$$A_1 = \{a_1, \bar{a}_1\} = \{\text{died}, \text{alive}\},$$
$$A_2 = \{a_2, \bar{a}_2\} = \{\text{smoker}, \text{nonsmoker}\}, \tag{18}$$
$$B = \{b, \bar{b}\} = \{\text{age } 18 - 64, \text{ age } 65 - 74\}, \tag{19}$$

where $p(\bar{b}) = 0.1334$, and where $b$ and $\bar{b}$ denote age-groups. According to the data of [16], Simpson's paradox reads [see Appendix D.1 for several technical clarifications]:

$$p(a_1|a_2) = 0.2214 < p(a_1|\bar{a}_2) = 0.2485, \tag{20}$$
$$p(a_1|a_2, b) > p(a_1|\bar{a}_2, b),$$
$$p(a_1|a_2, \bar{b}) > p(a_1|\bar{a}_2, \bar{b}). \tag{21}$$

Note that $B$ here influences $A_1$: the age of a person is a predictor of his/her survival. There are few people who quit or started smoking, so causal influences from $B$ to $A_2$ can be ignored [16]. We can assume that influences from smoking to age are absent. Then this example is intermediate between two situations considered in [7–9, 13]. Recall that when $B$ influenced $A_2$, these references advised to decide via the fine-grained option of the paradox, while for the case of the inverse influence (from $A_2$ to $B$) they recommend to employ the coarse-grained version; see (7, 8) and Fig. 1.

Hence, we should expand on the above situation to achieve a workable model. We can assume that $A_2$ and $B$ are influenced by a common cause. Genetic factors influence an individual's age and tendency to smoke. Originally proposed by Fisher [46], this hypothesis was later substantiated in several studies; see Refs. [47, 48] for reviews. Note that this refers to genetics of the smoking behavior itself, and not to health problems that can be caused by smoking plus genetic factors. Several sets of studies that contributed to genetic determinants of smoking behavior are as follows. *(i)* Children of smoking parents tend to smoke. *(ii)* Smoking behavior of adopted kids correlates stronger with that of their biological parents. *(iii)* Monozygotic (genetically identical) twins correlate in their smoking behavior much stronger than heterozygotic twins. Smoking behavior includes both the acquisition and maintenance of smoking. Monozygotic twins show correlations in both these aspects.

As a preliminary hypothesis, we suggest that genetic factors are the common cause of both smoking and age. To apply **Theorem 1** we introduce genetic variable $C = \{$risk to smoking, no risk to smoking$\}$. TODAG (14) can describe our simplified model. Now we need to consider genes, which can have influence nicotine addiction and show evidence of pleiotropy, i.e., they can influence more than one aspect of health and survival [49]. CHRNA5 is a pleiotropic gene that encodes subunits of the nicotinic acetylcholine receptor, which is important in neural signaling and nicotine addiction. The receptor can influence various aspects of smoking behavior: nicotine binding and response, reward pathways, craving intensity, smoking cessation success rates, *etc*; see Ref. [50] for a review. CHRNA5 has two alleles G (guanine) and A (adenine), which differ by a single nucleotide. Now A is the risk allele, which is associated with increased smoking. G is the non-risk allele [50]. A is the dominant allele with respect to G, and we treat CHRNA5 as binary genetic variable; see Appendix D for clarifications. Thus, **Theorem 1** applies and we conclude – consistently with other studies – that smoking is not beneficial for survival.

# 5   Example: COVID-19, Italy *versus* China

Here the COVID-19 death rates are compared in Italy and China [31, 51]. According to the data, aggregated death rates in Italy are higher than in China, but in each age group, the death rates are higher in China. More precisely,

$$A_1 = \{a_1, \bar{a}_1\} = \{\text{died}, \text{alive}\},$$
$$A_2 = \{a_2, \bar{a}_2\} = \{\text{China}, \text{Italy}\}, \tag{22}$$
$$B = \{b, \bar{b}\} = \{\text{age } 60 - 79, \text{ age } 80+\}, \tag{23}$$

where $p(a_1)$ is the death rate out of COVID-19, $p(B)$ is found from the number of positively tested people in each age group, $p(\bar{b}) = 0.1012$, and where $p(\bar{b}|a_2) = 0.1017$ and $p(\bar{b}|\bar{a}_2) = 0.3141$. According to the data of [31], Simpson's paradox reads

$$p(a_1|a_2) = 0.0608 < p(a_1|\bar{a}_2) = 0.0760, \tag{24}$$
$$p(a_1|a_2, b) = 0.0507 > p(a_1|\bar{a}_2, b) = 0.04900,$$
$$p(a_1|a_2, \bar{b}) = 0.150 > p(a_1|\bar{a}_2, \bar{b}) = 0.135. \tag{25}$$

The authors of [31] proposed that this situation is described by TODAG $A_2 \to B \to A_1 \leftarrow A_2$; cf. (8). Then the conclusion from [9, 13] will be that the aggregated version of Simpson's paradox works, i.e. Italy did worse than China. The authors of Ref. [31] reached the same conclusion.

When applying the common cause set-up from section 3.1, we can look at (12), because $A_2$ is better described as a label (avoiding dealing with the probability of country). Hence, from the viewpoint of (12), we need a common cause that supplements $A_2$ and acts on both $A_1$ and $B$. We propose that the quality of healthcare system can be the common cause $C$ here. In particular, a more affordable healthcare system may cause a higher proportion of older people in the country's society. Indeed, for 2019, Italy had a larger percentage of people aged above 65 than China: 24.05 % *versus* 12.06 %. On the other hand, the healthcare system will influence death rates in all age groups. If $C$ is binary, then our conclusion from **Theorem 1** is opposite to that of [31]: China did worse than Italy.

# 6   Simpson's paradox and common cause principle for Gaussian variables

## 6.1   Formulation of Simpson's paradox for continuous variables

Simpson's paradox is uncovered earlier for continuous variables than for the discrete case [1]. Researching the continuous variable paradox and identifying it in big datasets is currently an active research field [23, 24, 29, 52–54].

The association between continuous variables $A_1 = \{a_1\}$ and $A_2 = \{a_2\}$ can be based on a reasonable definition of correlation coefficient [1, 30]. We focus on Gaussian variables, because this definition is unique for them and amounts to conditional variance. These variables are also important in the context of machine learning (e.g. linear regressions) [55].

Hence the formulation of Simpson's paradox given $B = \{b\}$ reads instead of (1–3) [1, 23, 24, 30]:

$$\sigma[a_1, a_2]\sigma[a_1, a_2|b] < 0 \text{ for all } b, \tag{26}$$
$$\sigma[a_1, a_2] \equiv \langle (a_1 - \langle a_1 \rangle)(a_2 - \langle a_2 \rangle) \rangle, \tag{27}$$
$$\sigma[a_1, a_2|b] \equiv \langle (a_1 - \langle a_1 \rangle_b)(a_2 - \langle a_2 \rangle_b) \rangle_b, \quad \langle a \rangle_b \equiv \int \mathrm{d}a\, a\, p(a|b), \tag{28}$$

where $\langle a \rangle_b$ and $\sigma[a_1, a_2|b]$ are the conditional mean and covariance; $\langle a \rangle$ and $\sigma[a_1, a_2]$ are the mean and covariance; $p(a|b)$ is the conditional probability density of $A = \{a\}$.

The message of (26) is that the usual and conditional covariance have different signs, i.e., they predict different types of associations between $A_1$ and $A_2$. For instance, $\sigma[a_1, a_2] > 0$ means correlation, while $\sigma[a_1, a_2|b]$ implies anti-correlation. Note a subtle difference between this formulation of Simpson's paradox and that presented in section 2.2. In (26–27) the formulation is symmetric with respect to $A_1$ and $A_2$.

## 6.2 General solution for Gaussian variables

For fuller generality, we shall assume that $A = \{\boldsymbol{a}\}$, and $B = \{\boldsymbol{b}\}$ are Gaussian column vectors with a number of components (i.e., dimensionality) $n_A$, and $n_B$, respectively. We also define

$$\boldsymbol{y}^{\mathrm{T}} = (\boldsymbol{a}, \boldsymbol{b}), \quad \boldsymbol{y}^{\mathrm{T}}\boldsymbol{y} \text{ is a number}, \quad \boldsymbol{y}\boldsymbol{y}^{\mathrm{T}} \text{ is a matrix}, \tag{29}$$

where T means transposition. We assume that a Gaussian $n_X$-dimensional variable $X = \{\boldsymbol{x}\}$ is the common cause variable for $A$ and $B$:

$$P(\boldsymbol{y}|\boldsymbol{x}) = (2\pi)^{-n_A/2} \left(\det[\mathcal{Q}]\right)^{-1/2} e^{-\frac{1}{2}(\boldsymbol{y}-\mathcal{C}\boldsymbol{x})^{\mathrm{T}}\mathcal{Q}^{-1}(\boldsymbol{y}-\mathcal{C}\boldsymbol{x})}, \tag{30}$$

$$P(\boldsymbol{x}) = (2\pi)^{-n_X/2}(\det[\mathcal{S}])^{-1/2} e^{-\frac{1}{2}\boldsymbol{x}^{\mathrm{T}}\mathcal{S}^{-1}\boldsymbol{x}}, \tag{31}$$

$$\mathcal{Q} = \begin{pmatrix} \mathcal{A} & 0 \\ 0 & \mathcal{B} \end{pmatrix}, \qquad \langle\boldsymbol{y}\rangle_{\boldsymbol{x}} = \mathcal{C}\boldsymbol{x}, \tag{32}$$

$$\langle(\boldsymbol{y} - \langle\boldsymbol{y}\rangle_{\boldsymbol{x}})(\boldsymbol{y}^{\mathrm{T}} - \langle\boldsymbol{y}^{\mathrm{T}}\rangle_{\boldsymbol{x}})\rangle_{\boldsymbol{x}} = \mathcal{Q}, \tag{33}$$

where the common cause feature of $X = \{\boldsymbol{x}\}$ is ensured by the block-diagonal structure of the covariance matrix $\mathcal{Q}$: $\mathcal{A}$ and $\mathcal{B}$ are (resp.) covariance matrices for $A$ and $B$. In (30), $\mathcal{C}$ is $(n_A + n_B) \times n_X$ matrix that ensures the coupling between $(A, B)$ and $X$. For simplicity and without loss of generality we assumed that $\langle\boldsymbol{x}\rangle = 0$ and hence $\langle\boldsymbol{y}\rangle = 0$ in (30). We get from (30) after arranging similar terms (and omitting normalization):

$$P(\boldsymbol{x})P(\boldsymbol{y}|\boldsymbol{x}) \propto e^{-\frac{1}{2}[\boldsymbol{x}^{\mathrm{T}} - \boldsymbol{y}^{\mathrm{T}}\mathcal{Q}^{-1}\mathcal{C}V^{-1}]V[\boldsymbol{x} - V^{-1}\mathcal{C}^{\mathrm{T}}\mathcal{Q}^{-1}\boldsymbol{y}] - \frac{1}{2}\boldsymbol{y}^{\mathrm{T}}[\mathcal{Q}^{-1} - \mathcal{Q}^{-1}\mathcal{C}V^{-1}\mathcal{C}^{\mathrm{T}}\mathcal{Q}^{-1}]\boldsymbol{y}}, \tag{34}$$

$$V = \mathcal{S}^{-1} + \mathcal{C}^{\mathrm{T}}\mathcal{Q}^{-1}\mathcal{C}. \tag{35}$$

Employing (87) from Appendix F we obtain:

$$\mathcal{Q}^{-1} - \mathcal{Q}^{-1}\mathcal{C}V^{-1}\mathcal{C}^{\mathrm{T}}\mathcal{Q}^{-1} = (\mathcal{Q} + \mathcal{C}\mathcal{S}\mathcal{C}^{\mathrm{T}})^{-1}, \tag{36}$$

$$P(\boldsymbol{y}) \propto e^{-\frac{1}{2}\boldsymbol{y}^{\mathrm{T}}(\mathcal{Q}+\mathcal{C}\mathcal{S}\mathcal{C}^{\mathrm{T}})^{-1}\boldsymbol{y}}, \tag{37}$$

$$\langle\boldsymbol{y}\boldsymbol{y}^{\mathrm{T}}\rangle = \mathcal{Q} + \mathcal{C}\mathcal{S}\mathcal{C}^{\mathrm{T}}, \tag{38}$$

We now recall (29, 33), introduce the block-diagonal form for $\mathcal{C}\mathcal{S}\mathcal{C}^{\mathrm{T}}$, and find

$$\mathcal{Q} + \mathcal{C}\mathcal{S}\mathcal{C}^{\mathrm{T}} = \begin{pmatrix} \mathcal{A} + \mathcal{J} & \mathcal{K} \\ \mathcal{K}^{\mathrm{T}} & \mathcal{B} + \mathcal{L} \end{pmatrix}, \tag{39}$$

$$(\mathcal{Q} + \mathcal{C}\mathcal{S}\mathcal{C}^{\mathrm{T}})^{-1} = \begin{pmatrix} (\mathcal{A} + \mathcal{J} - \mathcal{K}(\mathcal{B} + \mathcal{L})^{-1}\mathcal{K}^{\mathrm{T}})^{-1} & \dots \\ \dots & \dots \end{pmatrix}, \tag{40}$$

where (40) is deduced via formulas from Appendix F. In (40) we need only the upper-left block, so that all other blocks are omitted. Collecting pertinent expressions from (29, 38, 40, 33), we obtain along with (38):

$$\langle(\boldsymbol{y} - \langle\boldsymbol{y}\rangle_{\boldsymbol{x}})(\boldsymbol{y}^{\mathrm{T}} - \langle\boldsymbol{y}^{\mathrm{T}}\rangle_{\boldsymbol{x}})\rangle_{\boldsymbol{x}} = \mathcal{Q}, \tag{41}$$

$$\langle(\boldsymbol{a} - \langle\boldsymbol{a}\rangle_{\boldsymbol{b}})(\boldsymbol{a}^{\mathrm{T}} - \langle\boldsymbol{a}^{\mathrm{T}}\rangle_{\boldsymbol{b}})\rangle_{\boldsymbol{b}} = \mathcal{A} + \mathcal{J} - \mathcal{K}(\mathcal{B} + \mathcal{L})^{-1}\mathcal{K}^{\mathrm{T}}. \tag{42}$$

## 6.3 The minimal set-up of Simpson's paradox: 3 scalar variables + scalar cause

For this simplest situation, $\boldsymbol{y}^{\mathrm{T}} = (a_1, a_2, b)$ is a 3-dimensional vector, $\mathcal{A}$ is a $2 \times 2$ matrix, $\mathcal{C}$ is a $3 \times 1$ matrix, while $\mathcal{S}$ and $\mathcal{B}$ are positive scalars. Now (38, 41, 42) read:

$$\langle(a_1 - \langle a_1\rangle_b)(a_2 - \langle a_2\rangle_b)\rangle_b = \mathcal{A}_{12} + \mathcal{C}_{11}\mathcal{C}_{21}\mathcal{S}\,\epsilon,$$

$$0 < \epsilon \equiv \frac{\mathcal{B}}{\mathcal{B} + \mathcal{C}_{31}^2\mathcal{S}} < 1, \tag{43}$$

$$\langle(a_1 - \langle a_1\rangle_x)(a_2 - \langle a_2\rangle_x)\rangle_x = \mathcal{A}_{12}, \tag{44}$$

$$\langle a_1 a_2\rangle = \mathcal{A}_{12} + \mathcal{C}_{11}\mathcal{C}_{21}\mathcal{S}. \tag{45}$$

Now consider a scenario of Simpson's paradox, where

$$\langle a_1 a_2\rangle = \mathcal{A}_{12} + \mathcal{C}_{11}\mathcal{C}_{21}\mathcal{S} > 0 \text{ and} \tag{46}$$

$$\langle(a_1 - \langle a_1\rangle_b)(a_2 - \langle a_2\rangle_b)\rangle_b = \mathcal{A}_{12} + \mathcal{C}_{11}\mathcal{C}_{21}\mathcal{S}\,\epsilon < 0. \tag{47}$$

Due to $0 < \epsilon < 1$, these two inequalities demand $\mathcal{A}_{12} < 0$. Likewise, $\langle a_1 a_2 \rangle = \mathcal{A}_{12} + \mathcal{C}_{11}\mathcal{C}_{21}\mathcal{S} < 0$ and $\mathcal{A}_{12} + \mathcal{C}_{11}\mathcal{C}_{21}\mathcal{S}\,\epsilon > 0$ demand $\mathcal{A}_{12} > 0$. It is seen that under Simpson's paradox for this minimal situation, the sign of $\langle (a_1 - \langle a_1 \rangle_x)(a_2 - \langle a_2 \rangle_x) \rangle_x$ coincides with the sign of $\langle (a_1 - \langle a_1 \rangle_b)(a_2 - \langle a_2 \rangle_b) \rangle_b$. We are thus led to the following:

**Theorem 2:** In the minimal situation (43–45) with the (minimal) common cause, the continuous Simpson's paradox (26) is resolved in the sense that the decision on the sign of correlations should proceed according to the fine-grained option: $\langle (a_1 - \langle a_1 \rangle_b)(a_2 - \langle a_2 \rangle_b) \rangle_b$; see (26–27).

For non-minimal common causes, all possibilities of the paradox can be realized; see Appendix G.

## 7    Conclusion

We addressed Simpson's paradox: the problem of setting up an association between two events $a_1$, $a_2$ given the lurking variable $B$. This decision-making paradox provides two plausible but opposite suggestions for the same situation; see (1) and (2, 3). Either the first option is correct, the second option is correct, or none of them is correct.

We focus on cases when there is a common cause $C$ for $B$ and $A = (A_1, A_2)$ (which combines $a_1$, $a_2$ and their complements). Alternatively, $C$ screens out $A$ from $B$; cf. Fig. 1. These cases include those in which there is no causal influence from $A$ to $B$, as well as from $B$ to $A$. Hence, the dependency between $A$ and $B$ are to be explained via the common cause $C$, which is a statement of the common cause principle [35, 36]. Now the association between $a_1$ and $a_2$ is to be decided by looking at $p(a_1 | a_2, c)$ for various values of $C$. This task is normally difficult given the fact that $C$ is frequently not fully known and is not observed. However, provided that $A_1$, $A_2$, $B$ and $C$ are binary, $p(a_1 | a_2, c)$ shows the same association as the option (2, 3) of Simpson's paradox. In this sense, Simpson's paradox is resolved in the binary situation, provided that the situation allows a binary cause or a binary screening variable. The same conclusion on resolving Simpson's paradox was reached for Gaussian variables in the minimal situation. Several examples can illustrate the plausibility of a minimal $C$.

Our solution of Simpson's paradox is not a generalization of the existing solution, since it employs a different idea. As we argued in section 2.2.2, the only unambiguous solution proposed so far refers to the directed acyclic graph (7). We also provided a counter-example against an opinion that the solution of Simpson's paradox can be decided based on the directed acyclic graph structure only.

We provide the first resolution of Simpson's paradox for Gaussian variables. This scenario of the paradox differs from the discrete in at least one essential aspect (it is symmetric), and was historically known earlier than the discrete version, and is more frequent in practice. This scenario of the paradox cannot be analyzed via standard directed acyclic graphs, and has to be worked out directly.

Our results have several limitations, but (we believe) these can be overcome with further research. *(i)* We limited ourselves to results that hold for all (minimal) common causes. For many applications, this is too stringent: if the common cause is known to exist, but is not observed directly, then it may be sufficient to infer it e.g. via the (generalized) maximum likelihood [45] or the minimal entropy method [44]. This may provide pertinent information on the real common cause and the structure of Simpson's paradox. *(ii)* We insisted on a precise common cause. The screening relation (10) is also useful, when it does hold approximately, but the support of $C$ is relatively small. Such an approximate relation (10) provides data compression via feature detection, which is the main message of unsupervised methods such as Non-negative Matrix factorization and Probabilistic Latent Dirichlet indexing [39, 40]. The impact of such approximate, but efficient causes on probabilistic reasoning is an interesting research subject that we plan to explore in the future. *(iii)* All examples we presented are observationally incomplete: a plausible cause had to be inferred, but it was not shown to exist or function from the real data.

### Acknowledgments and Disclosure of Funding

This work was supported by the HESC of Armenia under Grants 24FP-1F030, 21AG-1C038 and 21T-1C037.

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

# A How frequent is Simpson's paradox: an estimate based on the non-informative Dirichlet density

To estimate the frequency of Simpson's paradox under fair data-gathering, we can try to generate the probabilities in (1–3) randomly in an unbiased way, and calculate the frequency of holding the paradox [30, 56]. The best and widely accepted candidate for an unbiased density of probabilities is the Dirichlet density, which is widely employed in statistics and machine learning [57, 58]. The Dirichlet probability density for $n$ probabilities $(q_1, ..., q_n)$ reads:

$$\mathcal{D}(q_1, ..., q_n | \alpha_1, ..., \alpha_n) = \frac{\Gamma[\sum_{k=1}^{n} \alpha_k]}{\prod_{k=1}^{n} \Gamma[\alpha_k]} \prod_{k=1}^{n} q_k^{\alpha_k - 1} \, \delta(\sum_{k=1}^{n} q_k - 1), \tag{48}$$

$$\int_0^\infty \prod_{k=1}^{n} \mathrm{d}q_k \mathcal{D}(q_1, ..., q_n | \alpha_1, ..., \alpha_n) = 1, \tag{49}$$

where $\alpha_k > 0$ are the parameters of the Dirichlet density, $\delta(x)$ is the delta-function, and $\Gamma[x] = \int_0^\infty \mathrm{d}q \, q^{x-1} e^{-q}$ is the Euler's $\Gamma$-function. Since $\mathcal{D}(q_1, ..., q_n)$ is non-zero only for $q_k \geq 0$ and $\sum_{k=1}^{n} q_k = 1$, the continuous variables themselves have the meaning of probabilities.

Many standard prior densities for probabilities are contained in (48); e.g., homogeneous ($\alpha_1 = ... \alpha_n = 1$), Haldane's ($\alpha_1 = ... \alpha_n = \alpha \approx 0$), Jeffreys ($\alpha_1 = ... \alpha_n = 1/2$). For estimating the frequency of Simpson's paradox, Ref. [56] employed homogeneous and Jeffreys prior.

For modeling a non-informative Dirichlet density we find it natural to take

$$\alpha_1 = ... \alpha_n = 1/n. \tag{50}$$

The homogeneity feature, $\alpha_1 = ... \alpha_n$ in (50) is natural for an unbiased density. The factor $\frac{1}{n}$ in (50) makes an intuitive sense, since $\alpha_1 = ... \alpha_n$ become homogeneous (non-informative) probabilities. Eq. (50) arises when we assume that the distribution of random probabilities is independent of whether they were generated directly from (48) with $n$ components, or alternatively from (48) with $nm$ components $\alpha_1 = .... = \alpha_{nm}$, and then marginalized. This requirement indeed leads to (50), as can be checked with the following feature of (48):

$$\int_0^\infty \mathrm{d}q'_{n-1} \, \mathrm{d}q'_n \, \delta(q_{n-1} - q'_{n-1} - q'_n) \times$$
$$\mathcal{D}(q_1, ..., q_{n-2}, q'_{n-1}, q'_n | \alpha_1, ..., \alpha_n)$$
$$= \mathcal{D}(q_1, ..., q_{n-2}, q_{n-1} | \alpha_1, ..., \alpha_{n-2}, \alpha_{n-1} + \alpha_n). \tag{51}$$

The message of (51) is that aggregating over two probabilities leads to the same Dirichlet density with the sum of the corresponding weights $\alpha_{n-1}$ and $\alpha_n$.

We estimated the frequency of Simpson's paradox assuming that 8 probabilities $p(A_1, A_2, B)$ in (1–3) are generated from (48, 50) with $n = 8$ (binary situation). This amounts to checking two relations (they amount to (1–3) and its reversal)

$$[p(a_1|a_2) - p(a_1|\bar{a}_2)][p(a_1|a_2, b) - p(a_1|\bar{a}_2, b)] < 0, \tag{52}$$
$$[p(a_1|a_2, b) - p(a_1|\bar{a}_2, b)][p(a_1|a_2, \bar{b}) - p(a_1|\bar{a}_2, \bar{b})] > 0.$$

Our numerical result is that the frequency of two inequalities in (52) is $\approx 4.29\% \pm 0.001\%$. For this precision it was sufficient to generate $N = 10^7$ samples from (48, 50) with $n = 8$. This result compares favorably with $\approx 1.66\%$ obtained for $\alpha_1 = ... \alpha_8 = 1$ (homogeneous prior), and $\approx 2.67\%$ obtained for $\alpha_1 = ... \alpha_8 = 0.5$ (Jeffreys prior) [56]. It is seen that the frequency of Simpson's paradox is a decreasing function of $\alpha_1 = ... = \alpha_8 = \alpha$ [56].

Roughly, the above result $\approx 4.29\%$ means that in every 1000 instances of 3 binary variables, 42 instances will show Simpson's paradox. This number is reassuring: it is not very large meaning that the standard decision-making based on the marginal probabilities in (1) will frequently be reasonable. But it is also not very small, showing that Simpson's paradox is generic and has its range of applicability.

## B  Proof of Theorem 1

The main idea of proving (13) is inverting (9):

$$p(a_1|a_2, c)$$

$$= \frac{p(\bar{b}|\bar{c})p(a_1|a_2, b)p(b|a_2) + (p(\bar{b}|\bar{c}) - 1)p(a_1|a_2, \bar{b})p(\bar{b}|a_2)}{p(\bar{b}|\bar{c})p(b|a_2) + (p(\bar{b}|\bar{c}) - 1)p(\bar{b}|a_2)} \quad (53)$$

$$= p(a_1|a_2, \bar{b}) + \frac{p(\bar{b}|\bar{c})p(b|a_2)[p(a_1|a_2, \bar{b}) - p(a_1|a_2, b)]}{1 - p(\bar{b}|\bar{c}) - p(b|a_2)}, \quad (54)$$

$$p(c|a_2) = \frac{p(\bar{b}|\bar{c})p(b|a_2) + (p(\bar{b}|\bar{c}) - 1)p(\bar{b}|a_2)}{p(b|c) + p(\bar{b}|\bar{c}) - 1}$$

$$= \frac{p(\bar{b}|\bar{c}) + p(b|a_2) - 1}{p(b|c) + p(\bar{b}|\bar{c}) - 1}, \quad (55)$$

where unknown quantities $p(a_1|a_2, c)$ and $p(c|a_2)$ are represented via known ones (i.e. $p(A_1, A_2, B)$) and free parameters $p(B|C)$. Eqs. (54, 55) hold upon changing $a_2$ by $\bar{a}_2$ and are deduced in Appendix E via specific notations that should be useful when dealing with (9) for a non-binary $C$.

The rest of the proof is algebraic but non-trivial. It also works out and employs constraints (4, 64) on Simpson's paradox itself. Expanding both sides of (1),

$$p(a_1|a_2) = p(a_1|a_2, b)p(b|a_2) + p(a_1|a_2, \bar{b})p(\bar{b}|a_2), \quad (56)$$

$$p(a_1|\bar{a}_2) = p(a_1|\bar{a}_2, b)p(b|\bar{a}_2) + p(a_1|\bar{a}_2, \bar{b})p(\bar{b}|\bar{a}_2), \quad (57)$$

and using there (2, 3) we subtract the sides of (1) from each other and find:

$$p(a_1|a_2, b) + p(\bar{b}|a_2)[p(a_1|a_2, \bar{b}) - p(a_1|a_2, b)] <$$
$$p(a_1|\bar{a}_2, b) + p(\bar{b}|\bar{a}_2)[p(a_1|\bar{a}_2, \bar{b}) - p(a_1|\bar{a}_2, b)]. \quad (58)$$

We return to (2, 3) and note that we can assume without loosing generality

$$p(a_1|a_2, \bar{b}) > p(a_1|a_2, b). \quad (59)$$

Eqs. (56, 57) imply that for the validity of (1–3, 59) it is necessary to have $p(a_1|\bar{a}_2, \bar{b}) > p(a_1|a_2, b)$, which together with (2, 3, 59) revert to (4). Now (1, 56, 57) read

$$p(a_1|a_2, b) + [1 - p(b|a_2)](p(a_1|a_2, \bar{b}) - p(a_1|a_2, b)) \quad (60)$$
$$< p(a_1|\bar{a}_2, b) + [1 - p(b|\bar{a}_2)](p(a_1|\bar{a}_2, \bar{b}) - p(a_1|\bar{a}_2, b)),$$

$$p(a_1|a_2, \bar{b}) - p(b|a_2)(p(a_1|a_2, \bar{b}) - p(a_1|a_2, b))$$
$$< p(a_1|\bar{a}_2, \bar{b}) - p(b|\bar{a}_2)(p(a_1|\bar{a}_2, \bar{b}) - p(a_1|\bar{a}_2, b)), \quad (61)$$

where (60) and (61) are equivalent. Eqs. (60, 61, 4) imply

$$[1 - p(b|a_2)](p(a_1|a_2, \bar{b}) - p(a_1|a_2, b)) <$$
$$[1 - p(b|\bar{a}_2)](p(a_1|\bar{a}_2, \bar{b}) - p(a_1|\bar{a}_2, b)), \quad (62)$$

$$p(b|a_2)(p(a_1|a_2, \bar{b}) - p(a_1|a_2, b)) >$$
$$p(b|\bar{a}_2)(p(a_1|\bar{a}_2, \bar{b}) - p(a_1|\bar{a}_2, b)). \quad (63)$$

As checked directly, Eqs. (62, 63) lead to

$$p(b|a_2) > p(b|\bar{a}_2). \quad (64)$$

Now we return to (55) and assume there $p(b|c) + p(\bar{b}|\bar{c}) - 1 < 0$, which leads to $p(\bar{b}|\bar{c}) + p(b|a_2) - 1 < 0$ from (55). Writing down from (55) the formula for $p(c|\bar{a}_2)$ and making the same assumption we get $p(\bar{b}|\bar{c}) + p(b|\bar{a}_2) - 1 < 0$. Now look at (54) and its analog obtained via $a_2 \to \bar{a}_2$, and use there these two results together with (63, 64) and (4) to deduce the first inequality in (13) under assumption $p(b|c) + p(\bar{b}|\bar{c}) - 1 < 0$. It should be obvious that the second inequality in (13) holds under the same assumption since we nowhere used any specific feature of $c$ compared to $\bar{c}$.

For $p(\bar{b}|\bar{c}) + p(b|a_2) - 1 > 0$ we need to use instead of (54) another form of (53)

$$p(a_1|a_2, c) = p(a_1|a_2, b) -$$
$$\frac{[1 - p(\bar{b}|\bar{c})]p(\bar{b}|a_2)[p(a_1|a_2, b) - p(a_1|a_2, \bar{b})]}{p(\bar{b}|\bar{c}) + p(b|a_2) - 1}. \tag{65}$$

The rest is similar to the above: we proceed via (62, 64) and (4) and deduce (13) from (55), (65) and the analog of (65) obtained via $a_1 \to \bar{a}_2$.

## C  More examples of Simpson's paradox

We collected several examples of the paradox that are scattered in the literature. We discuss them employing our notations in equations (1–3) of the main text emphasizing (whenever relevant) the existence of the common cause (or screening) variable $C$.

**Example 5.** Snow tires provide cars with better traction in snowy and icy road conditions. However, nationally in the US, cars fitted with snow tires are more likely to have accidents in snowy and icy conditions [59]. $A_1 = \{a_1 = \text{accident}, \bar{a}_1 = \text{no} - \text{accident}\}$, $A_2 = \{a_2 = \text{changed tires}, \bar{a}_2 = \text{not changed}\}$, $B = \{\text{states}\}$. Here the choice of the state (warm or cold) has a direct causal link to accidents in winter conditions. Now snow tires tend to be fitted to cars only in snowy winter months and in states with colder weather. Cars in warmer months and in states with warmer weather are much less likely to have accidents in snowy and icy conditions. Plausibly, there is a random variable, $C = \{\text{good weather conditions}, \text{bad weather conditions}\}$, which causes $A$, and screens $A$ from $B$: $p(A|CB) = p(A|C)$. The times are distributed as $t_B < t_C < t_{A_2} < t_{A_1}$.

**Example 6.** This example emerged from discussing our own experience with hospitals. We need to choose between two hospitals 1 and 2: $A_1 = \{a_1 = \text{recovered}, \bar{a}_1 = \text{not recovered}\}$, $A_2 = \{a_2 = \text{hospital } 1, \bar{a}_2 = \text{hospital } 2\}$, $B = \{\text{first half} - \text{year}, \text{second half} - \text{year}\}$, $C = \{\text{types of illness}\}$. Here we do not expect direct causal influence from $B$ to $A$, if (as we assume) the hospitals do not treat seasonal illnesses. We expect that $C$ causes $A$, and screens it from $B$.

Note that the data from which the probabilities for Simpson's paradox are calculated is the number of patients $N(A_1, A_2, B)$ that came to the hospital. Simpson's paradox does not occur if within each season the hospitals accept an equal number of patients: $\sum_{A_1} N(A_1, A_2, B)$ does not depend on the value of $B$. This creates a conceptual possibility for judging between the hospitals. This is however not realistic, because imposing on these hospitals an equal number of patients can disturb their usual (normal) functioning.

**Example 7.** Simpson's paradox is realized when comparing scores of professional athletes, e.g. the batting averages of baseball players [60]. Here $A_1$ refers to a score of an athlete in a game, e.g. $A_1 = \text{high score}, \text{low score}$, $A_2$ denotes concrete athletes, while $B$ is the time-period (e.g. playing season). The causing variable $C$ can refer to the psychological and physical state of an athlete that influences his/her game success, and the number of games he/she participated in each season.

## D  Elaborations on smoking and surviving

**1.** Our interest in this subject started from learning about the works by R. Fisher, who proposed that at least a part of the association between smoking and survival may be due to genetic common causes. Then we noted that qualitative genetists clarified his statements in 1980s, but they are nearly forgotten in modern genetics, where genetic determinants of smoking are still studied actively [49].

The data presented in Ref. [16] considers three random variables $A_1 = \{\text{died, alive}\}$, $A_2 = \{\text{smoking, non-smoking}\}$, and $B = \{\text{younger, older}\}$. To this we added an unobserved genetic variable: $C = \{\text{risk to smoking, no risk to smoking}\}$, which roughly corresponds to the gene CHRNA5 described below.

A fairly general TODAG for this situation is

$$A_1 \leftarrow B \leftarrow C \to A_2 \to A_1, \quad A_2 \to B, \quad C \to A_1, \tag{66}$$

e.g. because once $C$ can influence $B$, then potentially also $A_2$ can have a direct influence on $B$. At the present stage of our knowledge on pertinent genetic and age-dependent factor influencing smoking, this TODAG is not manageable. So we had to simplify it drastically.

First, we erased the link $A_1 \leftarrow B$, because the physical age by itself does not influence survival. The physiological age correlates well with the physical age for some people, which can already affect their survival rate. However, the physiological age in this experiment was not recorded or controlled.

Once we assumed $A_1 \not\leftarrow B$, then from the viewpoint of Simpson's paradox, it was already natural to assume $A_2 \not\to B$ as well, because the link $A_2 \to B$ does not influence $p(A_1|\mathrm{do}(A_2))$. (Postulating the direct influences $A_2 \to B$ are more or less akin to predetermining the influences of smoking.) At any rate, we emphasize that both $A_1 \not\leftarrow B$ and $A_2 \not\to B$ are essential assumptions of the model. We these assumptions we end up from (66) with the following TODAG [cf. (14)]:

$$B \leftarrow C \to A_2 \to A_1, \quad C \to A_1. \tag{67}$$

It remains to explain in which sense the gene can influence the age. For example, if an allele of a gene (see below) can be a common cause of both smoking and (independently) smoking-generated deceases, then aged (but still healthy) people can be those which did not have this allele.

**2.** The classical Mendelian genetics assumed (and in many instanced verified) that as far the influence on the phenotype is concerned, one can restrict a gene to a binary variable: recessive and dominant alleles of the gene. Each organism has two genes (one from mother and another one from father), and now the three pairs - recessive-dominant, dominant-recessive, dominant-dominant - amount to one type of influence to the phenotype, while the version recessive-recessive to another influence.

In modern genetics, many exclusion from this classical binary-gene law are known. For example, the gene of the blood type has 3 alleles (A,B, and O). Here A and B are dominant with respect to O, but together they are co-dominant and hence there are 4 blood groups: AO, BO, AB, OO.

To understand whether the genes controlling the smoking behavior can be modeled as binary (i.e. dominant and recessive), we need to consider concrete genes, which according to current genetics research have serious effects on nicotine addiction and show evidence of pleiotropy, i.e., they can influence more than one aspect of health and survival; see [49].

CHRNA5 is a gene that encodes subunits of the nicotinic acetylcholine receptor, which is important in neural signaling and nicotine addiction. The receptor can influence various aspects of smocking behavior: nicotine binding and response, reward pathways, craving intensity, smoking cessation success rates, *etc*; see Ref. [50] for a review.

CHRNA5 has two alleles G and A. They are denoted by G (guanine) and A (adenine), because the alleles differ by single nucleotide. Now A is the risk allele, which is associated with increased smoking. G is the non-risk allele [50]. Now A is the dominant allele with respect to G, and CHRNA5 can be said to be binary with the following reservation: there is a dose-effect and the pair AA turns out to be more risky than AG (in contrast to GG, which is risk free). Within our crude model we neglect this difference and treat CHRNA5 as binary.

### D.1 Technical details on the example from section 4 of the main text

This example is taken from Ref. [16]. Its concise version was discussed in section 5. Here we provide more details on how the data was presented and how we analyzed it. In this case, binary $A_1$ represents the survival of a woman as determined by two surveys taken 20 years apart: $A_1 = \{\text{died}, \text{alive}\}$. The binary $A_2$ reads $A_2 = \{\text{smoker}, \text{nonsmoker}\}$, while $B = \{B_1, ..., B_6\}$ means the age group of the person recorded in the first survey. The $B_1$ now includes women between the ages of 18 and 24. Likewise, $B_2, B_3, B_4, B_5, B_6$ refer to (resp.) ages $(25-34)$, $(35-44)$, $(45-54)$, $(55-64)$, $(65-74)$. The corresponding probabilities read:

$$p(B_1) = 0.0946, \quad p(B_2) = 0.2272, \quad p(B_3) = 0.1859, \quad p(B_4) = 0.1681,$$
$$p(B_5) = 0.1908, \quad p(B_6) = 0.1334. \tag{68}$$

There is also the seventh age group that included people who were 75+ at the time of the first survey. We shall, however, disregard this group, since the data is pathological: nobody from this group survived till the second survey. It turns out that the aggregated data (1) of the main text hints that smoking is beneficial for survival:

$$p(A_1, A_2) = \sum_{k=1}^{6} p(A_1, A_2|B_k) p(B_k), \tag{69}$$

$$p(A_1 = \text{died}|A_2 = \text{smoking}) = 0.2214 < p(A_1 = \text{died}|A_2 = \text{nonsmoking}) = 0.2485. \tag{70}$$

This conclusion is partially reversed, once the age group $B$ is introduced:

$$p(A_1 = \text{died}|A_2 = \text{smoking}, B_k) > p(A_1 = \text{died}|A_2 = \text{nonsmoking}, B_k), \quad k = 1, 3, 4, 5, 6, \tag{71}$$

$$p(A_1 = \text{died}|A_2 = \text{smoking}, B_2) < p(A_1 = \text{died}|A_2 = \text{nonsmoking}, B_2). \tag{72}$$

We need to coarse-grain the above data to formulate the Simpson paradox clearly. Now

$$B = \{b, \bar{b}\}, \quad b = B_1 \cup B_2 \cup B_3 \cup B_4 \cup B_5, \qquad \bar{b} = B_6, \tag{73}$$
$$p(A_1 = \text{died}|A_2 = \text{smoking}, b) = 0.1820 > p(A_1 = \text{died}|A_2 = \text{nonsmoking}, b) = 0.1206, \tag{74}$$

$$p(A_1 = \text{died}|A_2 = \text{smoking}, \bar{b}) = 0.8056 > p(A_1 = \text{died}|A_2 = \text{nonsmoking}, \bar{b}) = 0.7829. \tag{75}$$

This leads to the formulation of Simpson's paradox discussed in section V of the main text. Eq. (73) is the only coarse-graining that leads to the paradox.

The authors of Ref. [16] provide the following heuristic explanation for the prediction difference between (70) and (71): they noted that aged people from the survey are mostly not smokers and most would have died out of natural reasons. This is the statistical explanation of the Simpson paradox. This explanation is not especially convincing because of (68, 72): it is seen that $B_2$ is the most probable group, for which (70) and (72) agree.

## E   Matrix notations for inverting the common cause equation

Here we develop matrix notations for inverting the common cause equation:

$$p(A_1, A_2, B) = \sum_C p(A_1, A_2, C)p(B|C), \tag{76}$$

where the summation goes over all values of $C$. We work for the case when the variables $A_1$, $A_2$, $B$ and $C$ are binary, though the matrix notations we introduce below are useful more generally.

Eq. (76) can be written in matrix form

$$\begin{pmatrix} [ik1] \\ [ik2] \end{pmatrix} = \begin{pmatrix} (1|1) & (1|2) \\ (2|1) & (2|2) \end{pmatrix} \begin{pmatrix} (ik1) \\ (ik2) \end{pmatrix}. \tag{77}$$

where $ik = 11, 12, 21, 22$ and the following notations were introduced

$$\begin{aligned}
&[111] \equiv p(a_1, a_2, b), \; [121] \equiv p(a_1, \bar{a}_2, b), \ldots, \\
&(111) \equiv p(a_1, a_2, c), \; (121) \equiv p(a_1, \bar{a}_2, c), \ldots, \\
&(1|1) \equiv p(b|c), \; (2|1) \equiv p(\bar{b}|c), \ldots, \\
&D = (2|2) + (1|1) - 1.
\end{aligned} \tag{78}$$

Inversion of the Eq. (77) gives

$$\begin{pmatrix} (ik1) \\ (ik2) \end{pmatrix} = \frac{1}{D} \begin{pmatrix} (2|2) & -(1|2) \\ -(2|1) & (1|1) \end{pmatrix} \begin{pmatrix} [ik1] \\ [ik2] \end{pmatrix}. \tag{79}$$

Eq. (79) implies

$$\begin{pmatrix} (i|k1)\{1|k\} \\ (i|k2)\{2|k\} \end{pmatrix} = \frac{1}{D} \begin{pmatrix} (2|2) & -(1|2) \\ -(2|1) & (1|1) \end{pmatrix} \begin{pmatrix} [i|k1][1|k] \\ [i|k2][2|k] \end{pmatrix}, \tag{80}$$

where analogously to (78) we introduced the following notations:

$$\begin{aligned}
&[1|11] \equiv p(a_1|a_2, b), \; [1|21] \equiv p(a_1, \bar{a}_2, b), \ldots, \\
&(1|11) \equiv p(a_1|a_2, c), \; (1|21) \equiv p(a_1|\bar{a}_2, c), \ldots, \\
&[1|1] \equiv p(b|a_2), \; [2|1] \equiv p(\bar{b}|a_2), \ldots, \\
&\{1|1\} \equiv p(c|a_2), \; \{2|1\} \equiv p(\bar{c}|a_2), \ldots.
\end{aligned} \tag{81}$$

The matrix relation (80) results in

$$(i|k1)\{1|k\} = \frac{(2|2)}{D}[i|k1][1|k] + \frac{(2|2)-1}{D}[i|k2][2|k]. \tag{82}$$

Using (82, 79) we get relations employed in the main text:

$$(i|k1) = \frac{(2|2)[i|k1][1|k]+((2|2)-1)[i|k2][2|k]}{(2|2)[1|k]+((2|2)-1)[2|k]}, \tag{83}$$

$$\{1|k\} = \frac{1}{D}(2|2)[1|k] + \frac{1}{D}((2|2))-1)[2|k], \tag{84}$$

$$(i|k2) = \frac{((1|1)-1)[i|k1][1|k]+(1|1)[i|k2][2|k]}{((1|1)-1)[1|k]+(1|1)[2|k]}, \tag{85}$$

$$\{2|k\} = \frac{1}{D}((1|1)-1)[1|k]] + [1|1][2|k]). \tag{86}$$

# F    Certain matrix relations

There is a useful formula for matrix inversion

$$(Z + UWV)^{-1} = Z^{-1} - Z^{-1}U(W^{-1} + VZ^{-1}U)^{-1}VZ^{-1}, \tag{87}$$

Eq. (87) is derived via two auxiliary formulas. First note that

$$V(1 + UV)^{-1} = (1 + VU)^{-1}V, \tag{88}$$

which follows from $(1 + UV)^{-1} = (V^{-1}(1 + VU)V)^{-1}$. Next, note moving $U$ according to (88)

$$U(1 + VU)^{-1}V = (1 + UV)^{-1}UV = 1 - (1 + UV)^{-1}, \tag{89}$$

which leads to

$$(1 + UV)^{-1} = 1 - U(1 + VU)^{-1}V. \tag{90}$$

To deduce (87) from (90), we manipulate $Z$ and $W$ in respectively LHS and RHS of (87), and hence transform (87) to the form (90), but with the following replacements: $U \to Z^{-1}U$ and $V \to WV$.

Eq. (87) leads to a generalized Sylvester formula:

$$\det[Z + UWV] = \det[Z]\det[W]\det[W^{-1} + VZ^{-1}U]. \tag{91}$$

The ordinary Sylvester formula for determinants reads

$$\det[I_{NN} - K_{NM}L_{MN}] = \det[I_{MM} - L_{MN}K_{NM}], \tag{92}$$

where $I_{NN}$ is the $N \times N$ unit matrix, $K_{NM}$ is a $N \times M$ matrix *etc.* Eq. (92) follows from the fact that (for $M \geq N$) $L_{MN}K_{NM}$ has the same eigenvalues as $K_{NM}L_{MN}$ (plus $M - N$ zero eigenvalues for $M - N > 0$).

Inverting a block matrix goes via

$$\begin{bmatrix} A_{11} & A_{12} \\ A_{21} & A_{22} \end{bmatrix}^{-1} = \begin{bmatrix} S^{-1} & -S^{-1}A_{12}A_{22}^{-1} \\ -A_{22}^{-1}A_{21}S^{-1} & A_{22}^{-1} + A_{22}^{-1}A_{21}S^{-1}A_{12}A_{22}^{-1} \end{bmatrix}, \tag{93}$$

$$S \equiv A_{11} - A_{12}A_{22}^{-1}A_{21}, \tag{94}$$

where dimensions of $A_{11}$, $A_{12}$, $A_{21}$ and $A_{22}$ are, respectively, $M \times M$, $M \times (N-M)$, $(N-M) \times M$, $(N - M) \times (N - M)$, and where $S$ is the Schur-complement of the block matrix over its upper diagonal part. Eq. (93) is straightforward to prove.

$$\det \begin{bmatrix} A_{11} & A_{12} \\ A_{21} & A_{22} \end{bmatrix} = \det[A_{11} - A_{12}A_{22}^{-1}A_{21}]\det[A_{22}]. \tag{95}$$

# G    Common cause with higher dimensionality for continuous variables

Let's discuss the scenario where the number of components of a common cause is two. Recall equation (55) of the main text and note that now $\mathcal{C}$ is a $3 \times 2$ matrix and $\mathcal{S}$ is a $2 \times 2$ matrix. For $\mathcal{C}\mathcal{S}\mathcal{C}^{\mathrm{T}}$ we have

$$\mathcal{C}\mathcal{S}\mathcal{C}^{\mathrm{T}} = \begin{bmatrix} v_1\mathcal{C}_{11} + v_2\mathcal{C}_{12} & v_1\mathcal{C}_{21} + v_2\mathcal{C}_{22} & v_1\mathcal{C}_{31} + v_2\mathcal{C}_{32} \\ v_1\mathcal{C}_{21} + v_2\mathcal{C}_{22} & u_1\mathcal{C}_{21} + u_2\mathcal{C}_{22} & u_1\mathcal{C}_{31} + u_2\mathcal{C}_{32} \\ v_1\mathcal{C}_{31} + v_2\mathcal{C}_{32} & u_1\mathcal{C}_{31} + u_2\mathcal{C}_{32} & k_1\mathcal{C}_{31} + k_2\mathcal{C}_{32} \end{bmatrix}, \tag{96}$$

where

$$v_1 = \mathcal{C}_{11}s_{11} + \mathcal{C}_{12}s_{21}, \quad v_2 = \mathcal{C}_{11}s_{12} + \mathcal{C}_{12}s_{22}, \tag{97}$$

$$u_1 = \mathcal{C}_{21}s_{11} + \mathcal{C}_{22}s_{21}, \quad u_2 = \mathcal{C}_{21}s_{12} + \mathcal{C}_{22}s_{22}, \tag{98}$$

$$k_1 = \mathcal{C}_{31}s_{11} + \mathcal{C}_{32}s_{21}, \quad k_2 = \mathcal{C}_{31}s_{12} + \mathcal{C}_{32}s_{22}. \tag{99}$$

We need to keep track of $_{12}$ element of the matrices, since

$$\langle(a_1 - \langle a_1\rangle_b)(a_2 - \langle a_2\rangle_b)\rangle_b = \left(\mathcal{A} + \mathcal{J} - \mathcal{K}(\mathcal{B} + \mathcal{L})^{-1}\mathcal{K}^{\mathrm{T}}\right)_{12}, \tag{100}$$

$$\langle(a_1 - \langle a_1\rangle_x)(a_2 - \langle a_2\rangle_x)\rangle_x = \mathcal{A}_{12}, \tag{101}$$

$$\langle a_1 a_2\rangle = (\mathcal{A} + \mathcal{J})_{12}, \tag{102}$$

and for which we have

$$(\mathcal{A} + \mathcal{J})_{12} = \mathcal{A}_{12} + v_1\mathcal{C}_{21} + v_2\mathcal{C}_{22}, \tag{103}$$

$$\left(\mathcal{K}(\mathcal{B} + \mathcal{J})^{-1}\mathcal{K}^{\mathrm{T}}\right)_{12} = \frac{1}{\mathcal{B} + k_1\mathcal{C}_{31} + k_2\mathcal{C}_{32}}(v_1\mathcal{C}_{31} + v_2\mathcal{C}_{32})(u_1\mathcal{C}_{31} + u_2\mathcal{C}_{32}). \tag{104}$$

Now, we consider the simplest case for a common cause

$$S = \begin{bmatrix} s & 0 \\ 0 & s \end{bmatrix}. \tag{105}$$

The equations simplify to

$$\langle(a_1 - \langle a_1\rangle_b)(a_2 - \langle a_2\rangle_b)\rangle_b = \mathcal{A}_{12} + s(\mathcal{C}_{11}\mathcal{C}_{21} + \mathcal{C}_{12}\mathcal{C}_{22}) \tag{106}$$

$$- \frac{s^2}{\mathcal{B} + s(c_{31}^2 + c_{32}^2)}(\mathcal{C}_{11}\mathcal{C}_{31} + \mathcal{C}_{12}\mathcal{C}_{32})(\mathcal{C}_{21}\mathcal{C}_{31} + \mathcal{C}_{22}\mathcal{C}_{32}), \tag{107}$$

$$\langle a_1 a_2\rangle = \mathcal{A}_{12} + s(\mathcal{C}_{11}\mathcal{C}_{21} + \mathcal{C}_{12}\mathcal{C}_{22}). \tag{108}$$

By setting $\mathcal{C}_{31} = 0$ and considering $s \gg \mathcal{B}$, we get

$$\langle(a_1 - \langle a_1\rangle_b)(a_2 - \langle a_2\rangle_b)\rangle_b = \mathcal{A}_{12} + s\mathcal{C}_{11}\mathcal{C}_{21}, \tag{109}$$

$$\langle a_1 a_2\rangle = \mathcal{A}_{12} + s(\mathcal{C}_{11}\mathcal{C}_{21} + \mathcal{C}_{12}\mathcal{C}_{22}). \tag{110}$$

Obviously, inequalities $\langle a_1 a_2\rangle > 0$ and $\langle(a_1 - \langle a_1\rangle_b)(a_2 - \langle a_2\rangle_b)\rangle_b < 0$ (or their inverted alternatives), do not determine the sign of $\mathcal{A}_{12}$, hereby the sign of $\langle(a_1 - \langle a_1\rangle_x)(a_2 - \langle a_2\rangle_x)\rangle_x$. Thus, for a common cause with two components we already see that it can support both fine-grained and coarse-grained options.

