# OpenReview forum: "Resolution of Simpson's paradox via the common cause principle"
_NeurIPS.cc/2025/Conference — NeurIPS 2025 poster_

### Official Review · Reviewer_Ffck · 2025-07-03

**Clarity:** 3
**Significance:** 2
**Originality:** 3
**Rating:** 4
**Confidence:** 2

**Summary:**

This paper revisits Simpson’s paradox and shows that it can be systematically resolved by assuming the existence of a minimal (possibly unobserved) common cause variable that renders the involved variables conditionally independent. The authors demonstrate that, in both binary and Gaussian settings, conditioning on such a common cause recovers consistent associations and resolves the paradox in its minimal form. They support their claim with examples from real-world data (e.g., smoking and COVID-19 cases) and provide theoretical results showing that decision-making based on fine-grained (conditioned) probabilities is valid when a minimal common cause exists.

**Questions:**

In the smoking and survival example, could the authors clarify the following:

- Why can the gene ($C$) be assumed to influence age ($B$) (Lines 234–243), while smoking behavior ($A_2$) is assumed to have no effect on age ($B$)?

- Why is it reasonable to model the gene ($C$) as a binary variable?

**Ethical Concerns:**

["NO or VERY MINOR ethics concerns only"]

**Final Justification:**

The author’s rebuttal has clarified my concern

**Limitations:**

yes

**Quality:**

3

**Strengths And Weaknesses:**

- The paper is well-written and fluent.
- It includes many examples that make the paper accessible.
-  The assumption of the existence of a common cause variable $C$ is quite reasonable, and the perspective of resolving Simpson's paradox through this common cause is novel and interesting.
- However, some of the additional assumptions about the common cause may not be realistic in real-world data, even in the examples discussed by the authors. Please see the questions for details.

---

> ### Author Rebuttal · Authors · 2025-07-31
>
> We thank the reviewer for evaluating our manuscript and asking two pertinent questions. Below we respond to them.
>
> Reviewer: Why can the gene $C$ be assumed to influence age $B$ (Lines 234-243), while smoking behavior $A_2$
> is assumed to have no effect on age $B$?
>
> Response: A fairly general DAG for this situation would be
> $A_1\leftarrow B\leftarrow C\to A_2\to A_1$ and $A_2\to B$, e.g. because
> once $C$ can influence $B$, then potentially also $A_2$ can have a
> direct influence on $B$.  At the present stage of our knowledge on pertinent
> genetic and age-dependent factor influencing smoking, this
> DAG is not manageable. So we had to simplify it drastically.
>
> First, we erased the link $A_1\leftarrow B$, because the physical age by itself does not influence survival. It is true that for some people, the physical age correlates well with the physiological age, which can already alter survival. However, the physiological age in this experiment was not recorded or controlled.
>
> Once we assumed $A_1\not\leftarrow B$, then from the viewpoint of
> Simpson's paradox it was already natural to assume $A_2\not\to B$ as
> well, because the link $A_2\to B$ does not influence $p(A_1|{\rm
> do}(A_2))$. At any rate, we emphasize that both $A_1\not\leftarrow B$
> and $A_2\not\to B$ are essential assumptions of the model.
>
> It remains to explain in which sense the gene can influence the age.
> For example, if an allele of a gene (see below) can be a common cause of
> both smoking and (independently) smoking-generated deceases, then aged (but healthy)
> people can be those which did not have this allele.
>
> Reviewer:
> Why is it reasonable to model the gene $C$ as a binary variable?
>
> Response: A given gene can have many alleles (i.e. versions), but the
> classical Mendelian genetics assumed (and in many instanced verified)
> that as far the influence on the phenotype is concerned, one can
> restrict the influence of a gene to the binary variable: recessive and
> dominant alleles. Each organism has two genes (one from mother and
> another one from father), and now the three pairs - recessive-dominant,
> dominant-recessive, dominant-dominant - amount to one type of influence
> to the phenotype, while the version recessive-recessive to another
> influence.
>
> In modern genetics, many exclusion from this classical binary-gene law
> are known. For example, the gene of the blood type has 3 alleles (A,B,
> and O). Here A and B are dominant with respect to O, but together they
> are co-dominant and hence there are 4 blood groups: AO, BO, AB, OO.
>
> To understand whether the genes controlling the smoking behavior can be
> modeled as binary (i.e. dominant and recessive), we need to consider
> concrete genes, which according to current genetics research have
> serious effects on nicotine addiction and show evidence of pleiotropy,
> i.e., they can influence more than one aspect of health and survival;
> please see The Tobacco and Genetics
> Consortium, "Genome-wide meta-analyses identify multiple loci associated
> with smoking behavior", Nat Genet. 42, 441-447 (2010).
>
> CHRNA5 is a gene that encodes subunits of the nicotinic acetylcholine
> receptor, which is important in neural signaling and nicotine addiction.  The
> receptor can influence various aspects of smocking behavior: nicotine
> binding and response, reward pathways, craving intensity, smoking
> cessation success rates, {\it etc}; please see for a review: G. Lassi,
> et al. "The CHRNA5–A3–B4 gene cluster and smoking: from discovery to
> therapeutics." Trends in neurosciences 39, 851-861 (2016).
>
> CHRNA5 has two alleles G and A.  They are denoted by G
> (guanine) and A (adenine), because the alleles differ by single
> nucleotide.  Now A is the risk allele, which is associated with
> increased smoking. G is the non-risk allele. Now A is the dominant
> allele with respect to G, and CHRNA5 can be said to be binary with the
> following reservation: there a dose-effect and the pair AA turns out to be
> more risky than AG (in contrast to GG, which is risk free). Within our
> crude model we neglect this difference and treat CHRNA5 as binary.

---

> ### Comment · Reviewer_Ffck · 2025-08-05
>
> Thank you for your efforts on rebuttal. I have raised the rating accordingly.

---

> > ### Author Response · Authors · 2025-08-05
> >
> > Thank you again for your useful questions.

---

### Official Review · Reviewer_GFht · 2025-07-03

**Clarity:** 4
**Significance:** 3
**Originality:** 3
**Rating:** 5
**Confidence:** 3

**Summary:**

This paper addresses the long-standing problem of Simpson’s paradox, where aggregated and disaggregated data lead to contradictory decisions. The authors propose resolving the paradox by assuming the existence of a common cause or screening variable, conditioned on which the relevant variables become independent. By conditioning on this (possibly unobserved) variable, the paradox can be resolved in two settings: (i) binary treatment, binary outcome and binary common cause, and a *minimal* setting with Gaussian variables. They show that if the common cause can take more than two values in the discrete case, it may not be possible to resolve the paradox unless the common cause is observed.

**Questions:**

1. Do you have practical guidelines for when it is reasonable to assume the existence of a common cause variable $C$, and when such an assumption might be implausible?

2. Related to the first question, in cases where the common cause is unobserved, how should one reason about or estimate its cardinality?

3. In the example of smoking and survival, could you expand on how your method would resolve the paradox? Specifically, given that genetic factors are unlikely to be binary, how would your approach handle this casE?

**Ethical Concerns:**

["NO or VERY MINOR ethics concerns only"]

**Final Justification:**

The authors answered most of my questions satisfactorily. Only one of my concerns remains, despite which I'd like to retain my positive assessment of the paper. I believe this paper introduces a nice perspective on Simpson's Paradox.

**Limitations:**

yes

**Paper Formatting Concerns:**

no concerns.

**Quality:**

3

**Strengths And Weaknesses:**

**Strengths**

1.Although the Simpson's paradox is a relatively old and well-studied problem, this paper provides a novel perspective on it.

2. The concrete real-world examples that were given were helpful and improved my understaning.

3. The paper is well-structured and has a clear presentation.

**Weaknesses**

Perhaps the main drawbacks of the paper are the following two:

1. Since the common cause is often unobserved, the criterion of Eq. (10) is not testable -- or at least if there are ways to falsify it, the authors do not mention anything in that regard.

2. The offered resolution only works if the common cause can take up to 2 values. This is also connected to the first weakness above, in the sense that an assumption must be made about an unobserved variable, which is not testable.

The authors acknowledge the second weakness, and discuss the limitation that if $C$ takes more than two values, the paradox may not be resolvable.

**Minor Comments**

- Equation (6) does not necessarily hold. I think the only point that the authors wanted to make was that $A_2$ and $B$ must be correlated, which is a true claim, but the argument for it does not have to go through Equation (6), for which counterexamples can be easily constructed.

- There are two typos that I think are important to fix: one is Equation (11), where either the right hand side must be multiplied with $p(c)$, or the first term should be the joint probability $p(A_1,A_2,c)$ rather than the conditional. The second one is on the first line of the proof of Theorem 1, where equations 51 and 52 are the results of inverting equation (11) rather than (10). Since the derivations appear much later in section E, this part was quite difficult to understand with the typo.

---

> ### Author Rebuttal · Authors · 2025-07-31
>
> We thank the reviewer for the clear and encouraging evaluation of our
> manuscript. We fully agree with all comments concerning misprints and
> unclarities.
>
> Response to weakness 1.
> When answering the question of the reviewer below, we shall explain in
> detail mechanisms of producing a common cause situation, even if it is
> not observed directly.  Here we would like to emphasize that there is a
> general consensus that resolving Simpson's paradox implies going beyond
> the three-variable data within which the paradox was formulated.  This
> ``looking beyond data'' is present in all {\it viable} solutions of
> Simpson's presented so far.
>
> By viable we mean the following. As reviewed in our manuscript,
> solutions to Simpson's paradox proposed so far separate into two groups.
> Within the first group (please our section 2.2.1) people proposed new
> ways of applying probability theory formulas. The proposals were to a
> large extent {\it ad hoc}, i.e. specially designed to solve the paradox.
> They were not tested more generally, and were not shown to lead to a
> fully coherent resolution schemes. For example, interchanging prediction
> with retrodiction (as proposed by Barigelli and Scozzafava) is certainly
> not a self-consistent thing to do and can lead to various errors when
> applied systematically.
>
> A viable solution was presented by Lindley and Novick and later on
> clarified and extended by Pearl; please see Refs.~[9,13] of our
> manuscript. It focuses on the three random variables $A_1$, $A_2$ and
> $B$, and demands that their mutual behavior is governed by a certain DAG
> (directed acyclic graph). There is no solution, before the structure of
> the DAG is clarified.
>
> Now the discussion we provided in section 2.2.2 illustrates that finding
> and motivating DAG is not easier than finding and motivating a common
> cause.
>
> Response to weakness 2. In this context,
> we would like to emphasize an important aspect of our theorems. For theorem
> 1, once we can motivate one of the above DAG with binary $C$, then no
> additional data-gathering is necessary for resolving the paradox, i.e.,
> we do not need to know $P(A_1,A_2|C)$, {\it etc}. Likewise, for theorem
> 2, once we assume the minimal Gaussian set-up, the paradox is resolved
> without knowing further details.
>
> If in the set-up of theorem 1, $C$ is not binary (but the common cause
> reasoning applies), then the resolution of the paradox is there in the
> sense that one should compare $p(a_1|a_2,C)$ with $p(a_1|\bar{a}_2,C)$.
> However, to understand the message of this resolution, one now needs the
> details of the common cause $C$. The same holds for theorem 2.
>
> Response to question 1. The assumption of common cause between
> $A=(A_1,A_2)$ and $B$ is reasonable if direct
> causal links between $A$ and $B$ are absent, i.e., when we suspect that the
> dependency between them is not causation. This situation is especially
> important for Simpson's paradox, because if $B$ causes $A_2$ and $A_1$,
> then it is intuitively expected that one should prefer the fine-grained
> version of Simpson's paradox (i.e. the version which conditions over
> $B$). This in fact was shown by Lindley and Novick, and Pearl within a
> DAG $A_1\leftarrow B\to A_2\to A_1$.
>
> A particular (but important) mechanism of producing a common cause is when $A$ and $B$
> are functions of $C$:
> $$
> A=f(C,\lambda), \qquad B=g(C,\mu),
> $$
> where $\lambda$ and $\mu$ are independent random variables: $p(\lambda,\mu)=p(\mu)p(\lambda)$.
> Now the common cause follows from
> $$
> p(a,b|c)=\sum_{\lambda,\mu} \delta(a-f(c,\lambda))\delta(b-g(c,\mu))p(\mu) p(\lambda)=p(a|c)p(b|c).
> $$
>
> Another important derivation of the common cause proceeds via the
> maximum entropy principle, when the probability $p(a,b,c)$ is recovered
> from maximizing the entropy $-\sum_{a,b,c}p(a,b,c)\ln p(a,b,c)$,
> assuming that $p(a,c)$ and $p(b|c)$ are known and are to be imposed as
> constraints in the maximization.
>
> Response to question 2. The estimation of the cardinality of the
> common cause is an important problem in linear algebra,
> matrix factorization theory, {\it etc}; please see J. E. Cohen and U. G.
> Rothblum, “Nonnegative ranks, decompositions, and factorizations of
> nonnegative matrices,” Linear Algebra and its Applications, 190,
> 149–168, (1993).
> N. Gillis, Nonnegative Matrix Factorization. SIAM, 2021.
>
>
> If the joint probability $p(a,b)$ of $A$ and $B$ is regarded as a matrix with non-negative elements, then common cause representations can be seen as factorizations of
> $p(a,b)$ into two matrices with non-negative elements:
> $$
> p(a,b)=\sum_{c=1}^{|C|} x_{ac}y_{cb}, \qquad x_{ac}\geq 0, \quad y_{cb}\geq 0,
> $$
> where the minimal value of $|C|$ (the minimal cardinality of the cause) is called the non-negative rank of the matrix $p(a,b)$.
> There exist efficient methods of estimating it; please see N. Gillis, Nonnegative Matrix Factorization. SIAM, 2021.
>
> Response to question 3.
>
> 1. Our interest in this subject started from learning about the
> works by R. Fisher, who proposed that at least a part of the association
> between smoking and survival may be due to genetic common causes.
> Then we noted that qualitative genetists clarified his statements in
> 1980s, but they are nearly forgotten in modern genetics, where genetic
> determinants of smoking are still studied actively; please see e.g., The
> Tobacco and Genetics Consortium, "Genome-wide meta-analyses identify
> multiple loci associated with smoking behavior", Nat. Genet. 42, 441-447
> (2010).
>
> The data presented in Ref.~[16] considers three random
> variables $A_1=$\{died, alive\}, $A_2=$\{smoking, non-smoking\}, and
> $B=$\{younger, older\}. To this, we added an unobserved genetic variable:
> $C=$\{risk to smoking, no risk to smoking\}, which roughly corresponds
> to the gene CHRNA5 described below. The real DAG that refers to these 4
> variables may be quite complicated. In particular, it may contain direct
> causal influences $A_2\to B$ in addition to more obvious relations $C\to
> B$ and $C\to A_2$. However, we also noted that postulating the direct
> influences $A_2\to B$ is more or less akin to predetermining the
> influences of smoking. Hence, we employed a simpler DAG $A_1\leftarrow C\to
> A_2\to A_1$, $C\to B$.
> Though the actual situation is probably much more complex, we thought it might be helpful to note that the simple DAG is already able to explain the data in a satisfactory way.
>
> More formally, a fairly general DAG for this situation would be
> $A_1\leftarrow B\leftarrow C\to A_2\to A_1$ and $A_2\to B$, e.g. because
> once $C$ can influence $B$, then potentially also $A_2$ can have a
> direct influence on $B$. At the present stage of our knowledge on pertinent
> genetic and age-dependent factors influencing smoking, this
> DAG is not manageable. So we had to simplify it drastically.
>
> First, we erased the link $A_1\leftarrow B$, because the physical age by itself does not influence survival. The physiological age correlates well with the physical age for some people, which can already affect their survival rate. However, the physiological age in this experiment was not recorded or controlled.
>
> Once we assumed $A_1\not\leftarrow B$, then from the viewpoint of
> Simpson's paradox, it was already natural to assume $A_2\not\to B$ as
> well, because the link $A_2\to B$ does not influence $p(A_1|{\rm
> do}(A_2))$. At any rate, we emphasize that both $A_1\not\leftarrow B$
> and $A_2\not\to B$ are essential assumptions of the model.
>
> It remains to explain in which sense the gene can influence the age. For example, if an allele of a gene (see below) can be a common cause of both smoking and (independently) smoking-generated deceases, then aged (but still healthy) people can be those who did not have this allele.
>
> 2. The classical Mendelian genetics assumed (and in many instances
> verified) that as far as the influence on the phenotype is concerned, one
> can restrict a gene to a binary variable: recessive
> and dominant alleles of the gene. Each organism has two genes (one from
> mother and another one from father), and now the three pairs -
> recessive-dominant, dominant-recessive, dominant-dominant - amount to
> one type of influence on the phenotype, while the version
> recessive-recessive to another influence.
>
> In modern genetics, many exclusions from this classical binary-gene law
> are known. For example, the gene of the blood type has 3 alleles (A,B,
> and O). Here, A and B are dominant with respect to O, but together they
> are co-dominant, and hence there are 4 blood groups: AO, BO, AB, OO.
>
> To understand whether the genes controlling smoking behavior can be
> modeled as binary (i.e., dominant and recessive), we need to consider
> concrete genes, which, according to current genetics research, have
> serious effects on nicotine addiction and show evidence of pleiotropy,
> i.e., they can influence more than one aspect of health and survival;
> please see The Tobacco and Genetics
> Consortium, "Genome-wide meta-analyses identify multiple loci associated
> with smoking behavior", Nat Genet. 42, 441-447 (2010).
>
> CHRNA5 is a gene that encodes subunits of the nicotinic acetylcholine
> receptor, which is important in neural signaling and nicotine addiction.  The
> receptor can influence various aspects of smoking behavior: nicotine
> binding and response, reward pathways, craving intensity, smoking
> cessation success rates, {\it etc}; please see for a review: G. Lassi,
> et al. "The CHRNA5–A3–B4 gene cluster and smoking: from discovery to
> therapeutics." Trends in neurosciences 39, 851-861 (2016).
>
> CHRNA5 has two alleles G and A.  They are denoted by G
> (guanine) and A (adenine), because the alleles differ by a single
> nucleotide.  Now, A is the risk allele, which is associated with
> increased smoking. G is the non-risk allele. Now A is the dominant
> allele with respect to G, and CHRNA5 can be said to be binary with the
> following reservation: there is a dose-effect, and the pair AA turns out to be
> more risky than AG (in contrast to GG, which is risk-free). Within our
> crude model, we neglect this difference and treat CHRNA5 as binary.

---

> > ### Comment · Reviewer_GFht · 2025-08-04
> >
> > I thank the authors for their detailed response. I am mostly satisfied with the answers except that I still find the binary $C$ in the example too much of an oversimplification. This is a bit concerning when one thinks of practical applicability, but given the negative results for the cases where this assumption is violated, I retain my positive assessment.

---

> > > ### Author Response · Authors · 2025-08-05
> > >
> > > We thank the reviewer for the second report. Below we respond to it.
> > >
> > > Reviewer: I thank the authors for their detailed response. I am mostly satisfied with the answers, except that I still find the binary $C$ in the example too much of an oversimplification. This is a bit concerning when one thinks of practical applicability, but given the negative results for the cases where this assumption is violated, I retain my positive assessment.
> > >
> > > Response: We agree that this is a limitation. However, the formalism we developed (both for discrete and Gaussian cases) allows one to study Simpson's paradox more generally, and possibly get further interesting results, e.g., about Simpson's resolvability of maximum entropy (most unbiased) common causes.

---

### Official Review · Reviewer_J6Fc · 2025-07-04

**Clarity:** 2
**Significance:** 2
**Originality:** 2
**Rating:** 4
**Confidence:** 3

**Summary:**

This paper claims to resolve Simpson's paradox through a new assumption of a common cause, but I think this is an insufficient description of their contributions.

Instead, the paper proposes a formulation of Simpson's paradox with a specific graphical structure, in which some $C$ is a common cause between the treatment, outcome, and a confounder.

In this setting, they claim that conditioning on the minimum common cause $B$ resolves the paradox.

**Questions:**

I am still having a difficult time understanding the criticism with the exchangeability resolution of Simpson's paradox. Is the problem just that TODAG's cannot express hidden confounding? Can't this be resolved using an ADMG and M-separation? There are still backdoor adjustments for ADMGs. If I am missing something here, I am happy to improve my score. However, as it stands, I must recommend rejection.

**Ethical Concerns:**

["NO or VERY MINOR ethics concerns only"]

**Final Justification:**

The authors clarified my misunderstanding about the goals of their paper. I think it is valuable to expand our understanding of the settings under which Simpson's paradox occurs. However, they do not provide a general resolution of the paradox, so I find the results somewhat incremental. This makes this a borderline accept.

**Limitations:**

yes

**Paper Formatting Concerns:**

no concerns

**Quality:**

2

**Strengths And Weaknesses:**

The paper is lacking the precision needed to really evaluate its results. As it stands, I am worried about correctness.
For example, they claim that some $C$ is a common cause for $A, B$. Since they then say that $C$ is unobserved, we are working with the following ADMG:

$A_1 \rightarrow A_2$

$A_1 \leftrightarrow A_2$

$A_1 \leftrightarrow B$

$A_2 \leftrightarrow B$

Conditioning on $B$ here should open up an active path between $A_1$ and $A_2$, so their claim seems to be incorrect with respect to Pearl's resolution. Their criticism of Pearl's resolution of the paradox seems to be that they believe the canonical setup $A_1 \rightarrow A_2$ with $A_1 \leftarrow B \rightarrow A_2$ does not describe all settings. But, of course, Pearl's resolution is more general and has to do with exchangeability. They do acknowledge this point, so there is a chance I am not understanding their problem with the exchangeability resolution. Conditioning on $B$ in their setting clearly does not ensure exchangeability.

Whether or not the paper is correct, the paper does not ``resolve'' Simpson's paradox in general. It simply studies the paradox in a specific setting.

Minor comment:
Line 140: It is a little confusing how you referred to Equations (1) and (2-3) so much later in the text. I think it might be easier to just refer to the aggregate direction vs the conditional one. Or use “Eq.” or “Equation.”

---

> ### Author Rebuttal · Authors · 2025-07-31
>
> We thank the reviewer for evaluating our manuscript. We are encouraged by
> the statement that the reviewer will be happy to improve the score, if
> we clarify our statements. We try to do so below.
>
> We identified 4 questions in the reviewer's report.
>
> **Q1** What precisely is shown in the present work?
>
> To answer this question it is easier to look at the following two DAGs;
> please see Fig.~1 for a condensed version of them.
> The first DAG is
> $B\leftarrow C\to A_2\to A_1$ and $C\to A_1$. The second DAG is $B\to
> C\to A_2\to A_1$ and $C\to A_1$.
>
> Now for the minimal set-up of binary variables Simpson's paradox leads to
> $$
> p(a_1|a_2,c)>p(a_1|\bar{a}_2,c), \quad p(a_1|a_2,\bar{c})>p(a_1|\bar{a}_2,\bar{c}),
> $$
> which in its turn produces
> $$
> p(a_1|{\rm do} (A_2=a_2))>p(a_1|{\rm do} (A_2=\bar{a}_2)).
> $$
> Altogether, it means Simpson's paradox is resolved in a specific way.
>
> **Q2** What precisely is our criticism of the Simpson's paradox solution presented by Pearl?
>
> In fact, our criticism consists of two separate points. The first one
> was presented in the manuscript, and will be reminded below. The second
> one is an expansion and formalization of one of criticisms presented by
> Armistead (our Ref.~[17]). We only very briefly mentioned it in the manuscript,
> so below we will provide its full details.
>
> A major part of Pearl's resolution (which expands on the previous work
> by Lindley and Novick; our Ref.~[13]) is that examples presented Lindley
> and Novick are mapped into two DAGs: $A_1\leftarrow B\to A_2\to A_1$ and
> $A_1\leftarrow A_2\to B\to A_1$. However, looking at those specific
> examples we found qualitative arguments for not restricting ourselves
> with those two DAGs only. For example, we motivate that the common cause
> situation might instead be a more plausible description of (at least)
> one of those examples. Such criticisms are not unexpected: any realistic
> example is (infinitely) complicated, and might admit different models.
>
> The second criticism is as follows. For the DAG $A_1\leftarrow A_2\to B\to A_1$, Pearl resolves the
> Simpson's paradox via $p(a_1|{\rm do} (A_2=a_2))$. However, for this specific DAG we have
> $$
> p(a_1|{\rm do} (A_2=a_2))=p(a_1|A_2=\bar{a}_2),
> $$
> and it is not clear what prevents us from going back to the original formulation of Simpson's paradox, i.e. what prevents us from
> comparing
> $$
> p(a_1|{\rm do} (A_2=a_2))<p(a_1|{\rm do} (A_2=\bar{a}_2)),
> $$
> with
> $$
> p(a_1|{\rm do} (A_2=a_2),b)>p(a_1|{\rm do} (A_2=\bar{a}_2),b), \quad
> p(a_1|{\rm do} (A_2=a_2),\bar{b})>p(a_1|{\rm do} (A_2=\bar{a}_2),\bar{b}).
> $$
> We understand that Lindley and Novick (despite of refraining to some extent from the causality language),
> and Pearl presented plausible arguments for resolving the paradox (for this DAG) via $p(a_1|{\rm do}( A_2=a_2))=p(a_1|A_2={a}_2)$.
> But still the level of resolution for the two DAGs is quite different, as was
> emphasized also by Armistead (our Ref.~[17]).
>
> **Q3** How Pearl's solution relates to exchangeability?
>
> To our understanding the exchangeability was proposed by Lindley and
> Novick as a partial substitute of the causality. Pearl (in his published
> papers and books) did not endorce the exchangeability viewpoint. This is
> also apparent from the review written by Lindley on the book by Pearl;
> please see D.V. Lindley, "Seeing and doing: the concept of causation",
> International Statistical Review, 70, 191-214 (2002).  After Lindley and
> Novick, the exchangeability viewpoint was extensively developed e.g. in
> O. Saarela, D.A. Stephens, and E.E. Moodie, "The role of exchangeability
> in causal inference", Statistical Science. 38, 369-385 (2023). This
> paper reviews the history of the probabilistic causal inference, and
> also explicitly supports the above opinion on Pearl and exchangeability.
>
> To our understanding, Pearl never formulated a general solution to
> Simpson's paradox. Following his general ideas, one can take $p(A_1|{\rm
> do} (A_2))$ as a possible general solution. As we stated explicitly when
> answering the first question, this solution agrees with our main
> result.
>
> **Q4** How do our results relate (and/or can be interpreted) via ADMGs.
>
> Consider again the two DAGs studied in our manuscript:
> $B\leftarrow C\to A_2\to A_1$ and $C\to A_1$; $B\to
> C\to A_2\to A_1$ and $C\to A_1$.
> Each of these DAGs leads to the following ADMG: $A_2\leftrightarrow A_1$,
> $A_2\leftrightarrow B$, $A_1\leftrightarrow B$, and $A_2\to A_1$. (Please note that the description of this ADMG by the reviewer contains a misprint.) Likewise, if one starts with this ADMG and assumes that there is only one hidden variable $C$, then one ends with one of the above DAGs. We see that $B$ is not a collider in the DAGs.
> We do not employ this ADMG in our consideration because the concept of m-separation here is not as useful, as working directly with the above two DAGs; please see our Fig.1.

---

> > ### Comment · Reviewer_J6Fc · 2025-08-03
> >
> > Thank you to the authors for this rebuttal. I will confirm a few things
> >
> > I believe I may have misunderstood the paper to be "assuming" the cases to be the last two DAGs shown in Figure 1. Is it, perhaps, more correct to say that the paper is "extending" the graph-based explanation of the paradox to those potential cases? My initial concern was that the paper was claiming that previous discussions of the first two DAGs were wrong and that all cases were covered by the last two DAGs.
> >
> > The ADMG you have given (I had switched my indices) does have $A_1 \leftrightarrow B \leftrightarrow A_2$. This is a collider, but since you are specifically studying just those two graphs in Figure 1, you are not considering the general case where $C$ is confounding all three. Is this correct? If so, then I agree there is no error and thank the reviewers for clearing that up.
> >
> > Finally, to clarify my point about Pearl's resolution: while Pearl does not use the terminology of exchangeability, this notion is shown to be equivalent to Pearl's graphical models by Richardson and Robins' work on SWIGs from 2013.
> >
> > In general, I take Pearl's resolution to be about whether $B$ forms a valid adjustment set for $A_1 | do(a_2)$. If you expand the DAGs in the last two examples of Figure 1 to include the $A_2 \rightarrow A_1$ edges, I believe the question again boils down to whether $B$ is a valid adjustment set. For this reason, I am still slightly skeptical about a "case-based" resolution, rather than the more general resolution using adjustment sets.

---

> > > ### Author Response · Authors · 2025-08-05
> > >
> > > We thank the reviewer for the second report. It is good to see that our opinions converge, and our work definitely benefited from the detailed criticisms of the reviewer. Below we respond to the reviewer's comments.
> > >
> > > Reviewer: I believe I may have misunderstood the paper to be "assuming" the cases to be the last two DAGs shown in Figure 1. Is it, perhaps, more correct to say that the paper is "extending" the graph-based explanation of the paradox to those potential cases?
> > >
> > > Response: We agree that our theorem 1 provides an extension of Simpson's paradox resolution to 2 new DAGs. Our theorem 2 treats Gaussian variables, and to some extent goes beyond DAGs, because Simpson's paradox for Gaussian variables (which historically came earlier than the discrete version; please see our section 6) is normally not formulated via DAGs.
> > >
> > > Reviewer: My initial concern was that the paper was claiming that previous discussions of the first two DAGs were wrong and that all cases were covered by the last two DAGs.
> > >
> > > Response: We did not say that the previous analysis was wrong. We said that perhaps the examples treated admit different models (those described by our DAGs).
> > >
> > > Reviewer: The ADMG you have given (I had switched my indices) does have $A_1 \leftrightarrow B \leftrightarrow A_2$. This is a collider, but since you are specifically studying just those two graphs in Figure 1, you are not considering the general case where $C$ is confounding all three. Is this correct? If so, then I agree there is no error and thank the reviewers for clearing that up.
> > >
> > > Response: We agree that we are studying specifically two DAGs: $B\leftarrow C\to A_2\to A_1$ and $C\to A_1$, and $B\to C\to A_2\to A_1$ and $C\to A_1$. The setup of theorem 1 applies to both these DAGs.
> > > If we consider the above ADMG: $A_2\leftrightarrow A_1$, $A_2\leftrightarrow B$, $A_1\leftrightarrow B$, $A_2\to A_1$, and require there that $C$ is the single and sole confounding variable with respect to all $\leftrightarrow$ present in the ADMG, then we end up with the first DAG: $B\leftarrow C\to A_2\to A_1$ and $C\to A_1$. In neither of the two DAGs studied by us, $B$ is a collider.
> > >
> > > Reviewer: To clarify my point about Pearl's resolution: while Pearl does not use the terminology of exchangeability, this notion is shown to be equivalent to Pearl's graphical models by Richardson and Robins' work on SWIGs from 2013.
> > >
> > > Response: Thank you for directing our attention to this pertinent reference, which tries to synthesize different approaches towards causality. It is possible that the general resolution of the paradox will be achieved within this synthesis.
> > >
> > > Reviewer: I take Pearl's resolution to be about whether $B$ forms a valid adjustment set for $A_1 | do(a_2)$. If you expand the DAGs in the last two examples of Figure 1 to include the $A_2 \rightarrow A_1$ edges, I believe the question again boils down to whether $B$ is a valid adjustment set.
> > >
> > > Response: The reviewer suggests the following general resolution of Simpson's paradox: take the fine-grained (i.e., conditioned over $B$) version of the paradox if $B$ is an adjustment variable for $A_1 | do(a_2)$. Take the coarse-grained (i.e., marginalized over $B$) version of the paradox if $B$ is not an adjustment variable. For clarity, we recall the standard definition of the non-adjustment variable for $A_1 | do(a_2)$: a collider ($...\to B\leftarrow ...$), or a mediator ($A_2\to B\to A_1$), or a descendant: $A_1\to B$. We agree with the reviewer that this suggestion is precisely what Lindley and Novick, and later Pearl, advised for the two DAGs: for $A_1\leftarrow B\to A_2\to A_1$, $B$ is not an adjustment variable, so we should take the fine-grained option. For $A_1\leftarrow A_2\to B\to A_1$, $B$ is a mediator, so we should take the coarse-grained version. Our theorem 1 is consistent with this resolution, since the setup of this theorem is based on one of two DAGs: $B\leftarrow C\to A_2\to A_1$ and $C\to A_1$ ; $B\to C\to A_2\to A_1$ and $C\to A_1$. For both cases, $B$ is an adjustment variable.
> > >
> > > However, we have a counterexample to the general applicability of this resolution. Consider the DAG $B\leftarrow C\to A_2\to A_1$ and $C\to A_1$, but now $C$ is not a binary variable: it can assume 3 values. Then we constructed numerical examples, where [please see our discussion after Eq.(14)]: $p(a_1|a_2, C) < p(a_1|\bar{a}_2, C)$, for $C=c_1,c_2,c_3$.  Now we got the coarse-grained version of the paradox, though for the considered DAG, $B$ is still an adjustment variable. For the tertiary common cause $C$, we also have $p(a_1|{\rm do}(A_2=a_2)) < p(a_1|{\rm do}(A_2=\bar{a}_2))$.
> > >
> > > Reviewer: I am still slightly skeptical about a "case-based" resolution, rather than the more general resolution using adjustment sets.
> > >
> > > Response: We clarified why looking at concrete graphs and situations for resolving Simpson's paradox is unavoidable. We do support the reviewer's opinion that Simpson's paradox must be solved generally.

---

> > > > ### Comment · Reviewer_J6Fc · 2025-08-07
> > > >
> > > > Given that the authors have clarified my initial misunderstanding about the contributions of the paper, I am revising my review to a 4: borderline accept. I feel that the paper makes a nice expansion of the cases in which Simpson's paradox can occur. However, since we still do not have a general solution, I am not sure the result is strong enough for a 5. I thank the authors for their rebuttal and discussion.

---

### Official Review · Reviewer_y8Yk · 2025-07-05

**Clarity:** 3
**Significance:** 2
**Originality:** 3
**Rating:** 4
**Confidence:** 3

**Summary:**

This paper studies Simpson’s paradox from a causal perspective and applies the result to decision-making with examples. The work first formalizes Simpson’s paradox. Consider a target event $A_1$ and an action $A_2$ performed before $A_1$, along with a context variable $B$ that also precedes $A_1$ (all variables are binary). The marginal distribution $p(a_1 \mid a_2) < p(a_1 \mid \bar{a}_2)$ can exhibit a different property than the conditional distribution $p(a_1 \mid a_2, b) > p(a_1 \mid \bar{a}_2, b)$, making it unclear whether to apply the action $A_2 = a_2$ or $A_2 = a_2'$.

Next, the paper reviews earlier attempts to resolve Simpson’s paradox and discusses their limitations. It then proposes a new approach: assume there exists a binary variable $C$ that is a cause of $A_1$ and $A_2$, and that the causal structure among $A_1$, $A_2$, $B$, and $C$ satisfies the condition that no additional information about $A_1$ and $A_2$ can be obtained from $B$ once we condition on $C$. Under this assumption, we have $p(a_1 \mid a_2, c) > p(a_1 \mid \bar{a}_2, c)$ and $p(a_1 \mid a_2, \bar{c}) > p(a_1 \mid \bar{a}_2, \bar{c})$, which aligns with the conditional distribution given $B$, and we should choose the action $A_2 = a_2'$. The result is further extended to the case of Gaussian variables.

**Questions:**

**Q1**. Is the term $P(s)$ missing in Equation (11)?

**Ethical Concerns:**

["NO or VERY MINOR ethics concerns only"]

**Final Justification:**

The paper presents a novel and sound solution to Simpson’s paradox under certain assumed graphical structures. The theoretical results offer interesting insights. However, I find the binary C assumption, as well as the graph over B and C, to be somewhat unnatural and difficult to validate. Therefore, I recommend a borderline accept.

**Limitations:**

Please refer to **W1** and **W2**.

**Quality:**

2

**Strengths And Weaknesses:**

# Strength

**S1.** The paper is well written. It provides a wide range of illustrative examples to support its claims and theorems, helping to ground the theoretical content in concrete scenarios. The logical flow is smooth and easy to follow.

**S2.** The theoretical results are presented with commendable mathematical rigor.

**S3.** The theoretical results are interesting in math.

# Weakness
**W1.**
A key concern lies in the use of Theorem 1 for decision-making. Theorem 1 states that conditioning on $B$ yields the same sign of association between $A_1$ and $A_2$ as conditioning on $C$, i.e.,

$$
p(a_1 \mid a_2, B) > p(a_1 \mid \bar{a}_2, B) \quad \text{then} \quad p(a_1 \mid a_2, C) > p(a_1 \mid \bar{a}_2, C).
$$

However, it is questionable whether this implies that taking action $A_2 = a_2$ is better than taking action $A_2 = \bar{a}_2$.

In causal inference, observational distributions reflect correlations, not causal effects. Decision-making should be based on causal effects, not merely on associations. As shown in \cite{pearl2009causality} (referenced as [13] in the paper), causal effects are defined in terms of the interventional distribution $p(a_1 \mid \text{do}(A_2 = a_2))$. It is entirely possible for the observational association to be positive while the causal effect is negative. For instance, in the presence of unobserved confounding between $A_1$ and $A_2$ (as in the bow arc graph in \cite{pearl2009causality}), it could happen that
$$
p(a_1 \mid \text{do}(a_2)) < p(a_1 \mid \text{do}(\bar{a}_2)),
$$
even though
$$
p(a_1 \mid a_2, C) > p(a_1 \mid \bar{a}_2, C).
$$
Therefore, the connection between Theorem 1 and decision-making should be more explicitly clarified, possibly by incorporating additional assumptions or causal identifiability arguments.

**W2.**
Another limitation of Theorem 1 lies in its strong assumptions. Although $C$ may be unobserved, the theorem relies on two difficult-to-verify assumptions: the conditional independence condition (Eq.~(10)) and the assumption that $C$ is binary. Since $C$ is unobserved, neither assumption can be verified from data, making the theorem challenging to apply in practice easily.

The paper acknowledges this limitation, stating:
“All examples we presented are observationally incomplete: a plausible cause had to be inferred, but it was not shown to exist or function from the real data.” (lines 337--338)

Given the considerable gap between the assumptions of the theorem and real-world data, and the lack of discussion on when or why these assumptions might be satisfied, I believe the practical contribution of Theorem 1 is limited.

---

> ### Author Rebuttal · Authors · 2025-07-31
>
> We thank the reviewer for the evaluation of our manuscript and for
> requesting important clarifications. The reviewer noted two weaknesses **W1**
> and **W2**, and asked one question **Q1**. Below we respond to them.
> **W1** and **Q1** can be addressed fully. **W2** cannot be
> addressed fully without serious additional research. Still, in the
> context of **W2** we presented a new argument in support of the
> publishability of our contribution.
>
> Response to **Q1**. Yes, indeed, $P(c)$ is missing in Equation (11).
>
> Response to **W1**. Our result is fully consistent with the
> do-calculus and with the proposal of the reviewer. To show this,
> consider two DAGs that support the causal structure of our theorems;
> please see Fig.~1 of the manuscript. (Please note that Fig. 1 wrote
> these two DAGs together in a condensed form, and also did not specify
> the causal direction $A_2\to A_1$.)
>
> The first DAG from Fig. 1 is
> $$
> B\leftarrow C\to A_2\to A_1, \qquad C\to A_1.
> $$
> The second is
> $$
> B\to C\to A_2\to A_1, \qquad C\to A_1.
> $$
> For both these DAGs
> we have
> $$
> p(a_1|{\rm do} (A_2=a_2) )=\sum_{C=c,\bar{c}} p(a_1|a_2,C) p(C).
> $$
> Hence, if (as stated in our theorem 1)
> $$
> p(a_1|a_2,c)>p(a_1|\bar{a}_2,c), \quad p(a_1|a_2,\bar{c})>p(a_1|\bar{a}_2,\bar{c}),
> $$
> then we get (as desired)
> $$
> p(a_1|{\rm do} (A_2=a_2))>p(a_1|{\rm do} (A_2=\bar{a}_2)).
> $$
>
> Response to **W2** consists of two parts.  In part **1** we argue
> that also the existing solutions of Simpson's paradox require conditions
> on data, and that our results compare well with the existing benchmarks.
> In part **2**, we discuss an important aspect of our theorems, which went
> unnoticed in the above criticism of the reviewer. For
>
> **1.** We agree that our results on resolving Simpson's
> paradox demand motivating one of the above DAGs. However, the
> viable resolution of the paradox (proposed by Lindley and Novick, and
> extended and formalized by Pearl; please see Refs.~[9,13] of our
> manuscript), also demands motivating certain DAGs.  This is the general
> point of Simpson's paradox: it is resolved only by going beyond its
> original, purely probabilistic formulation. More precisely, Simpson's
> paradox was so far unambiguously solved only for one DAG. To this we add
> two more DAGs outlined above. Hence we suggest that our contribution in
> resolving Simpson's is at least comparable to the existing
> ``benchmark''.
>
> Let us provide details of our opinion.
> Pearl in Ref.~[13] studied two DAGs. The first one is $A_1\leftarrow B\to A_2\to A_1$. For this DAG we agree that
> $$
> p(a_1|{\rm do} (A_2=a_2))>p(a_1|{\rm do} (A_2=\bar{a}_2))
> $$
> resolves the paradox. The resolution amounts to the fine-grained option of the paradox. This is expected: within
> $A_1\leftarrow B\to A_2\to A_1$, $B$ influences both $A_1$ and $A_2$; hence it is natural to condition over $B$.
>
> Pearl also considers another DAG: $A_1\leftarrow A_2\to B\to A_1$. For this situation
> $$
> p(a_1|{\rm do} (A_2))=p(a_1|A_2),
> $$
> and it is not clear what prevents us from going back to the original formulation of Simpson's paradox, i.e. what prevents us from
> comparing
> $$
> p(a_1|{\rm do} (A_2=a_2))<p(a_1|{\rm do} (A_2=\bar{a}_2)),
> $$
>
> with
>
> $$p(a_1|{\rm do} (A_2=a_2),b)>p(a_1|{\rm do} (A_2=\bar{a}_2),b),$$
> $$ p(a_1|{\rm do} (A_2=a_2),\bar{b})>p(a_1|{\rm do} (A_2=\bar{a}_2),\bar{b}).$$
>
> We understand that Lindley and Novick (despite of refraining to some extent from the causality language),
> and Pearl presented heuristic arguments for resolving the paradox also for the situation of the second DAG
> $A_1\leftarrow A_2\to B\to A_1$. But still the formal level of resolution for the two DAGs is quite different, as was
> emphasized also by Armistead (our Ref.~[17]).
>
> (Please note that the third DAG $B\leftarrow A_2\to A_1\to B$ is
> not studied, since we do not consider Simpson's paradox
> in the scenario where the variable $B$ is the effect of the involved variables $A_2$ and
> $A_1$.)
>
> To summarize, we see that so far Simpson's paradox was unambiguously
> solved only for one (out of two) DAGs.
>
> **2.** We would like to emphasize an important aspect of our theorems,
> which went unnoticed in the above criticism of the reviewer. For theorem
> 1, once we can motivate one of the above DAG with binary $C$, then no
> additional data-gathering is necessary for resolving the paradox, i.e.,
> we do not need to know $P(A_1,A_2|C)$,  etc. Likewise, for theorem
> 2, once we assume the minimal Gaussian set-up, the paradox is resolved
> without knowing further details.
>
> If in the set-up of theorem 1, $C$ is not binary (but one of the above
> DAGs applies), then the resolution of the paradox still applies in the
> sense that one should compare $p(a_1|a_2,C)$ with $p(a_1|\bar{a}_2,C)$,
> but to understand the message of this resolution, one now needs the
> details of the common cause $C$. The same holds for theorem 2.

---

> ### Comment · Reviewer_y8Yk · 2025-08-03
>
> Thanks for the detailed rebuttal provided by the reviewer. I have some follow-up discussions and comments regarding W1 and W2.
>
> **W1**. I agree that the result in the paper is consistent with do-calculus for the **two presented graphs in the rebuttal**. However, I would like to confirm whether there is an implicit assumption that **no additional confounders exist between $A_1$ and $A_2$**. If such confounders were present, I don’t believe the inequality $P(a_1 \mid do(a_2)) > P(a_1 \mid do(\bar{a}_2))$ would hold. I’m not opposed to this assumption, as it is also assumed in prior work such as [9]. However, I recommend making this assumption explicit in the paper, especially in Figure 1. While the first two graphs clearly reflect this assumption, the third and fourth graphs do not, which could lead to confusion about whether the assumption is required in those cases as well.
>
> **W2**. I agree with the main argument: the authors provide a resolution to the paradox that is at least comparable to existing “benchmark” approaches. In my opinion, the work offers interesting and novel insights into resolving the paradox, though it is not necessarily strictly superior to existing benchmarks.
>
> Additionally, I acknowledge the point that “once we can motivate one of the above DAGs with binary $C$, then no further data collection is needed to resolve the paradox.” However, my previous concern remains: since $C$ is unobserved, it may be practically more difficult to recognize the existence of such a binary variable. Perhaps the best way to justify this assumption is through prior expert knowledge?

---

> > ### Author Response · Authors · 2025-08-05
> >
> > We thank the reviewer for further comments about our results. Below we respond to those comments.
> >
> > Reviewer: W1. I agree that the result in the paper is consistent with do-calculus for the two presented graphs in the rebuttal. However, I would like to confirm whether there is an implicit assumption that no additional confounders exist between $A_1$ and $A_2$. If such confounders were present, I don’t believe the inequality $P(a_1 \mid do(a_2)) > P(a_1 \mid do(\bar{a}_2))$ would hold. I’m not opposed to this assumption, as it is also assumed in prior work such as [9]. However, I recommend making this assumption explicit in the paper, especially in Figure 1. While the first two graphs clearly reflect this assumption, the third and fourth graphs do not, which could lead to confusion about whether the assumption is required in those cases as well.
> >
> > Response: We fully agree with this suggestion and will certainly implement it in the revised version.
> >
> > Reviewer: W2. I agree with the main argument: the authors provide a resolution to the paradox that is at least comparable to existing “benchmark” approaches. In my opinion, the work offers new insights into resolving the paradox, though it is not necessarily strictly superior to existing benchmarks.
> >
> > Response: We agree with this point.
> >
> > Reviewer: Additionally, I acknowledge the point that “once we can motivate one of the above DAGs with binary $C$, then no further data collection is needed to resolve the paradox.” However, my previous concern remains: since $C$ is unobserved, it may be practically more difficult to recognize the existence of such a binary variable. Perhaps the best way to justify this assumption is through prior expert knowledge?
> >
> > Response: Yes, prior expert knowledge is the main source of such conclusions. We, in fact, applied it when working out our examples.

---

> > > ### Comment · Reviewer_y8Yk · 2025-08-08
> > >
> > > Thanks for the clarification. I will slightly raise my score to support the paper, given the clarification about the assumptions in the graph. However, I will not raise it to 5, as the unobserved binary variable C is very difficult to validate in practice.

---

### Note · Authors · 2025-08-12

We thank the editors and reviewers again for their consideration of our paper. We aim to highlight several strengths in our paper that either went unnoticed during the reviewer exchanges or were overshadowed by various technical details.

-- We provide the first resolution of Simpson's paradox for Gaussian variables. This scenario of the paradox differs from the discrete case in at least one essential aspect (it is symmetric), and is more frequent in practice. This scenario cannot be studied using standard DAGs (directed acyclic graphs) and must be analyzed directly.

-- As was acknowledged by one of the reviewers (but not all of them), our resolution of the paradox is fully comparable to the existing benchmarks provided by Lindley and Novick, and later on by Pearl. We add the following argument to this fact: all existing resolutions of the paradox (including ours) assume certain DAGs. However, the three-variable DAGs, which were assumed so far (by Lindley and Novick, and later more formally by Pearl), are not determined by the underlying joint probability; that is, they cannot be reconstructed solely from the joint probability. In such DAGs, additional assumptions about influence mechanisms are required beyond the proper probabilistic inference. This point was especially stressed by Lindley and Novick. Our assumption on DAGs is different, because our DAGs can be determined by the joint probability.

-- Besides providing two theorems on the resolution of Simpson's paradox, we develop formalisms (for the discrete and Gaussian versions of the paradox) that can address more general features of Simpson's paradox, e.g., its structure for inferred common causes. We anticipate that further interesting results will be obtained using our formalisms.

---

### Decision · Program_Chairs · 2025-09-17

**Decision:**

Accept (poster)

**Comment:**

**(a) Summary**
The paper addresses Simpson’s paradox by assuming a minimal common cause, presenting the first resolution for Gaussian variables and extending known DAG-based cases. The work combines rigorous theorems with illustrative examples and situates itself relative to classical results by Lindley, Novick, and Pearl.

**(b) Strengths**
- Novel treatment of Simpson’s paradox for Gaussian settings.
- Clear exposition supported by examples.
- Theoretical results are rigorous and connect to established causal inference literature.
- Author rebuttal successfully clarified key misunderstandings.

**(c) Weaknesses**
- Relies on strong assumptions, such as an unobserved binary common cause that may be difficult to justify in practice.
- Contributions extend benchmarks but do not provide a fully general solution.
- Some modeling simplifications may limit applicability.

**(d) Key reasons for decision**
Reviewers initially had mixed views, with concerns about correctness and assumptions. After rebuttal and discussion, reviewers **y8Yk** and **J6Fc** revised their ratings upward from reject/borderline reject to borderline accept, citing clarified contributions. Reviewers **GFht** and **Ffck** remained supportive throughout. Overall, the Gaussian case and DAG extensions were recognized as meaningful advances, even if not general.

**(e) Discussion and changes**
The discussion phase was productive: authors clarified alignment with do-calculus, addressed ADMG and exchangeability questions, and responded to practical concerns about modeling assumptions. While limitations remain, the reviewers’ final justifications reflect that these are acceptable trade-offs for a contribution of this scope.

**Decision: Accept (Poster).**
Concise justification: The paper provides rigorous and novel extensions to Simpson’s paradox, especially for Gaussian settings, and reviewers’ concerns were largely resolved in discussion.